# Spatially selective manipulation of cells with single-beam acoustical tweezers

Michael Baudoin [1,2✉], Jean-Louis Thomas[3], Roudy Al Sahely[1], Jean-Claude Gerbedoen[1], Zhixiong Gong [1], Aude Sivery[1], Olivier Bou Matar[1], Nikolay Smagin [1], Peter Favreau[1] & Alexis Vlandas [1✉]

Acoustical tweezers open major prospects in microbiology for cells and microorganisms contactless manipulation, organization and mechanical properties testing since they are biocompatible, label-free and have the potential to exert forces several orders of magnitude larger than their optical counterpart at equivalent power. Yet, these perspectives have so far been hindered by the absence of spatial selectivity of existing acoustical tweezers - i.e., the ability to select and move objects individually - and/or their limited resolution restricting their use to large particle manipulation only and/or finally the limited forces that they could apply. Here, we report precise selective manipulation and positioning of individual human cells in a standard microscopy environment with trapping forces up to ~200 pN without altering their viability. These results are obtained with miniaturized acoustical tweezers combining holography with active materials to synthesize specific wavefields called focused acoustical vortices designed to produce stiff localized traps with reduced acoustic power.

[1] Univ. Lille, CNRS, Centrale Lille, Yncréa ISEN, Univ. Polytechnique Hauts-de-France, UMR 8520—IEMN, SATT NORD, 59000 Lille, France. [2] Institut Universitaire de France, 1 rue Descartes, 75005 Paris, France. [3] Sorbonne Université, CNRS, Institut des NanoSciences de Paris, INSP, 75005 Paris, France. ✉email: michael.baudoin@univ-lille.fr; alexis.vlandas@iemn.fr

Contactless tweezers based on optical[1–3] and magnetic forces[4–6] have been developed in the last decades and have led to tremendous progress in science recognized by several Nobel prizes. Nevertheless, these technologies have stringent limitations when operating on biological matter. Optical tweezers rely on the optical radiation pressure, a force proportional to the intensity of the wavefield divided by the speed of light. The high value of the latter severely limits the forces which can be applied and imposes the use of high intensity fields. This can lead to deleterious photothermal damages (due to absorption-induced heating) and/or photochemical damages (due to excitation of reactive compounds like singlet oxygen)[7–10] adversely affecting cells' integrity. Magnetic tweezers, on the other hand, can only manipulate objects susceptible to magnetic fields and thus require other particles to be pre-tagged with magnetic compounds, a limiting factor for many applications. In microbiology, acoustical tweezers have many assets to become a prominent technology[11–15]. Indeed, they rely on the acoustical radiation force[15,16], which is—as for their optical counterpart—proportional to the intensity of the wave divided by the wave speed. But, the drastically lower speed of sound compared to light leads to driving power several orders of magnitude smaller than in optics to apply the same forces (or conversely, forces several orders of magnitude larger at the same driving power)[15,17,18]. In addition, the innocuity of ultrasounds on cells and tissues below cavitation and deleterious heating thresholds defined by the mechanical and thermal indexes is largely documented[19–23] and demonstrated daily by their widespread use in medical imaging[24]. Indeed, the frequencies typically used in medical ultrasound (1–100 MHz) and in the present work (~45 MHz) are far below electronic or molecular excitation resonances thus avoiding adverse effects on cells' integrity. In addition, the attenuation in water at these frequencies remains weak for manipulation at the micrometric scale, hence limiting absorption-induced thermal heating[22,23]. Finally, almost any type of particles (solid particles, biological tissues, drops) can be trapped without pre-tagging[14] and the low speed of sound enables spatial resolution down to micrometric scales even at these comparatively low frequencies.

Nevertheless, the promising capabilities offered by acoustical tweezers have so far been hindered by the lack of selectivity of existing devices[25,26] and/or their restricted operating frequency[17,27–31] limiting their use to large particles only. Here, selectivity refers to spatial selectivity, i.e. the ability to select and move an object independently of other neighboring objects. Yet, the ability to select, move and organize individual microscopic living organisms is of the utmost importance in microbiology for fields at the forefront of current research such as single-cell analysis, cell–cell interaction study, or to promote the emergence of disruptive research e.g. on spatially organized co-cultures. In this paper, we unleash the potential of acoustical tweezers by demonstrating individual biological cells' manipulation and organization in a standard microscopy environment with miniaturized single-beam acoustical tweezers. The strength and efficiency of acoustical tweezers is illustrated by exerting forces (~200 pN) on cells one order of magnitude larger than the maximum forces reported with optical tweezers[32], obtained with one order of magnitude less wave power (<2 mW). Cells' viability was assessed following exposure to the acoustic field measured by short- and long-term fluorescence viability assays.

## Results

### Acoustical tweezers' design.
First experimental evidences of large particles trapping with acoustic waves date back to the early twentieth century[33]. Nevertheless, the first demonstration of controlled manipulation of micrometric particles and cells with acoustic waves appeared only one century later with the emergence of microfluidics and high frequency transducers based on interdigitated electrodes[25,26]. In these recent works, trapping relies on the 2D superposition of orthogonal plane standing waves, an efficient solution for the collective motion of particles, but one which precludes any selectivity, i.e., the ability to select and move one particle out of a population[15]. Indeed, the multiplicity of nodes and antinodes leads to the existence of multiple trapping sites[34] which cannot be moved independently. In addition, multiple transducers or reflectors positioned around the manipulation area are mandatory for the synthesis of standing waves, a condition difficult to fulfill in many experimental configurations. With such orthogonal standing wave devices, Guo et al.[35] demonstrated (i) particles' collection at the multiple nodes of the standing wavefield; (ii) cells' patterning by bringing the cells one by one and waiting for each manipulated cell to adhere on the substrate in between two cells' manipulation (otherwise multiple free cells would move collectively and follow the same trajectory preventing their organization owing to the absence of spatial selectivity) and (iii) displacement of particles and cells along an axis perpendicular to the substrate by tuning the acoustic power to adjust the equilibrium between upward acoustic forces (acoustic radiation force and acoustic streaming) and downward gravity.

Selective trapping on the other hand requires strong spatial localization and hence tight focusing of the wavefield. In optics, this ability has been achieved with focused progressive waves[1], a solution also investigated in acoustics[36]. But such wavefields are inadequate in acoustics for most particles of practical interest, since objects with positive contrast factors (such as rigid particles or cells) are generally expelled from the focal point of a focused wave[37]. Acoustical vortices[38] provide an elegant solution to this problem[39]. These focused helical progressive waves spin around a central axis wherein the pressure amplitude vanishes, surrounded by a ring of high pressure intensity, which pushes particles toward the central node. Two-dimensional trapping[29,40] and three-dimensional levitation[27] and trapping[17] have been previously reported at the center of laterally and spherically focused vortices, respectively. Compared to tweezers based on focused beams operating in the Mie regime[41], the vortex-based tweezers enable to trap objects with positive contrast factors at the beam center, in 3D, and at lower operating frequencies, hence limiting deleterious heating. Conversely, these lower frequencies (and hence wavelength) lead to weaker gradients compared to tweezers operating in the Mie regime. However, all demonstrations with vortex-based tweezers were performed on relatively large particles (>300 μm in diameter) using complex arrays of transducers, which are cumbersome, not compatible with standard microscopes, and which cannot be easily miniaturized to trap micrometric particles. Recently, Baudoin et al.[30] demonstrated the selective manipulation of 150 μm particles in a standard microscopy environment with flat, easily integrable, miniaturized tweezers. To reach this goal, they sputtered holographic electrodes at the surface of an active piezoelectric substrate, designed to synthesize a spherically focused acoustical vortex.

Nevertheless, transcending the limits of this technology to achieve selective cells' manipulation remained a major scientific and technological challenge. Indeed, the system should be scaled down (frequency upscaling) by a factor of 10 (since cells have a typical size of 10 μm), while increasing drastically the field intensity, owing to the low acoustic contrast (density, compressibility) between cells and the surrounding liquid[42,43]. In addition, since the concomitant system's miniaturization and power increase are known to adversely increase the sources of dissipation, the tweezers had to be specifically designed to prevent

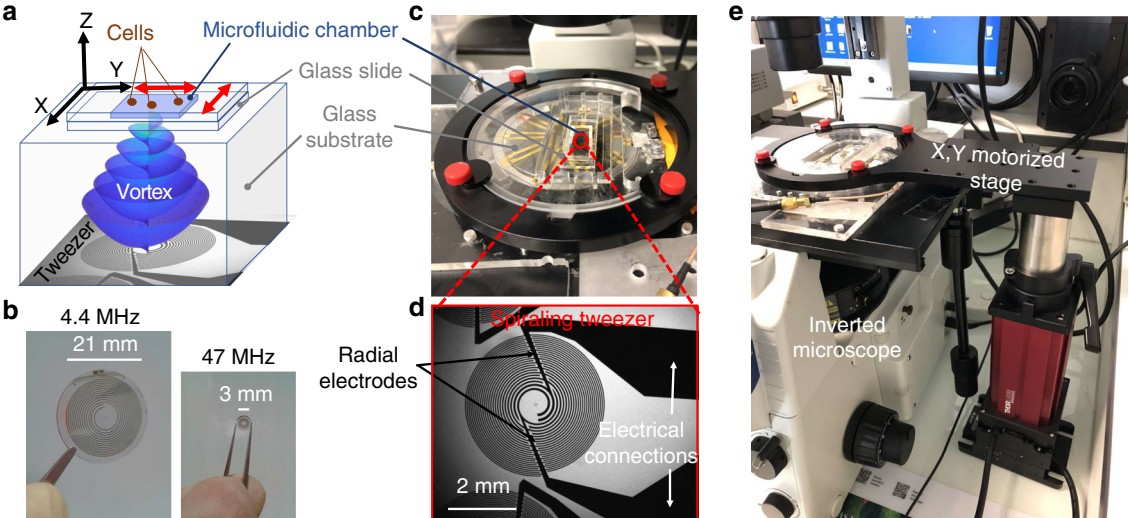

**Fig. 1 Experimental setup. a** Illustration of the working principle of the tweezers designed for cells' selective manipulation: A spherically focused acoustical vortex is synthesized by spiraling active electrodes metallized at the surface of a piezoelectric substrate and actuated with a function generator connected to an amplifier. The vortex propagates and focalizes inside a glued glass substrate and then reaches a microfluidic chamber made of a glass slide and a PDMS cover containing cells embedded in a growth medium. The microfluidic device is acoustically coupled with the transducer with a thin layer of silicone oil (25 cSt). A cell located at the center of the acoustical vortex is trapped. Its motion relative to other cells is enabled by the displacement of the microfluidic chamber driven by a XY motorized stage (see Supplementary Movie 1 for an animated explanation of the setup working principle). **b** Picture of typical transducers used in the present study (right) and illustration of the scale reduction compared to previous lower frequency designs by Baudoin et al.[30] (left). **c** Image of the actual experimental setup. **d** Zoom-in on the spiral transducer and the electrical connections (in black). **e** Illustration of the integration of the whole setup inside a standard inverted microscope. Photo credit: B: J.-C. Gerbedoen, SATT NORD/C-D-E: R.A. Sahely, Univ. Lille.

detrimental temperature increase and enable damage-free manipulation of cells.

First, spherically focused acoustical vortices (Fig. 1a) were chosen to trap the particles. Indeed, the energy concentration resulting from the 3D focalization (Fig. 2f) enables to reach high amplitudes at the focus from low power transducers. These spherically focused vortices were synthesized by materializing the hologram of a ~45 MHz vortex[30] with metallic electrodes at the surface of an active piezoelectric substrate. The hologram was discretized on two levels resulting in two intertwined spiraling electrodes (Fig. 1d), patterned in a clean room by standard photolithography techniques (see "Methods", section "Tweezers' design"). The scale reduction compared to our previous generation of acoustical tweezers[30] is illustrated in Fig. 1b. Second, the design of the electrodes was optimized to reduce Joule heating (magnified by the scale reduction) inside the electrodes. To prevent this effect, (i) the thickness of the metallic electrodes was increased by a factor of 2 (400 nm of gold and 40 nm of titanium); (ii) the width of the electrical connections (Fig. 1d) supplying the power to the spirals was significantly increased to prevent any dissipation before the active region; and (iii) two radial electrodes spanning half of the spirals were added as a way to effectively bring power to the driving electrode. Third, a 1.1 mm glass substrate (Fig. 1a, c) was glued to the electrodes and placed in between the transducers and the microfluidic chamber wherein the cells are manipulated. This glass substrate has a double function: (i) it enables the focalization of the wave and (ii) it thermally insulates the microfluidic device from the electrodes.

The final device hence consists of (see Supplementary Movie 1 in Supplementary Information, Fig. 1a, c, e): (i) spiraling holographic transducers excited with a sinusoidal electrical signal generating an acoustical vortex that propagates and focuses inside a glass substrate; (ii) a microfluidic PDMS chamber supported by a glass slide containing cells and placed on top of the substrate, wherein the acoustical vortex creates a trap and (iii) a motorized stage that enables the X,Y displacement of the microfluidic

chamber with respect to the trap. The whole transparent setup is integrated in an inverted microscope as depicted in Fig. 1e.

**Characterization of the acoustical trap.** The principle of high frequency acoustical vortices synthesis with these active holograms was assessed through the comparison of numerical predictions obtained from an angular spectrum code and experimental measurements of the acoustic field normal displacement at the surface of the glass slide (XY plane) with a Polytech UHF-120 laser Doppler vibrometer (Fig. 2a–d). Both the magnitude and phase are faithful to the simulations and demonstrate the ability to generate high frequency acoustic vortices. As expected, the wavefield exhibits a central node (corresponding to the phase central singularity) surrounded by a ring of high intensity which constitutes the acoustical trap. The magnitude of the sinusoidal acoustic field (displacement) depends on the driving electrical power and was measured to vary typically between 0.1 and 1 nm, at the electrical power used in the manipulation experiments. This corresponds to acoustic powers lying between 20 μW and 2 mW (see "Methods" section "Estimation of the acoustic power"). The concentration of the acoustic energy through focalization in the propagation plane (XZ) can be seen in Fig. 2e.

An estimation of the lateral force field exerted on a cell of 10 μm radius with density 1100 kg m$^{-3}$ and compressibility $4 \times 10^{-10}$ Pa$^{-1}$ was computed at each point in the manipulation plane of the microfluidic chamber (XY plane, Fig. 2f) with the theoretical formula derived by Sapozhnikov and Bailey[44]. This calculation gives an estimation of the force of the order of 100 pN, which can nevertheless strongly vary depending on the (unknown) cells' exact properties (see "Methods" section "Simulations of the acoustic field and radiation force" for the exact values for an acoustic vibration of 1 nm depending on cells' acoustic properties[42,43]). This order of magnitude agrees with the maximum force measured experimentally (200 pN) for similar parameters (see next subsection). These simulations of the lateral

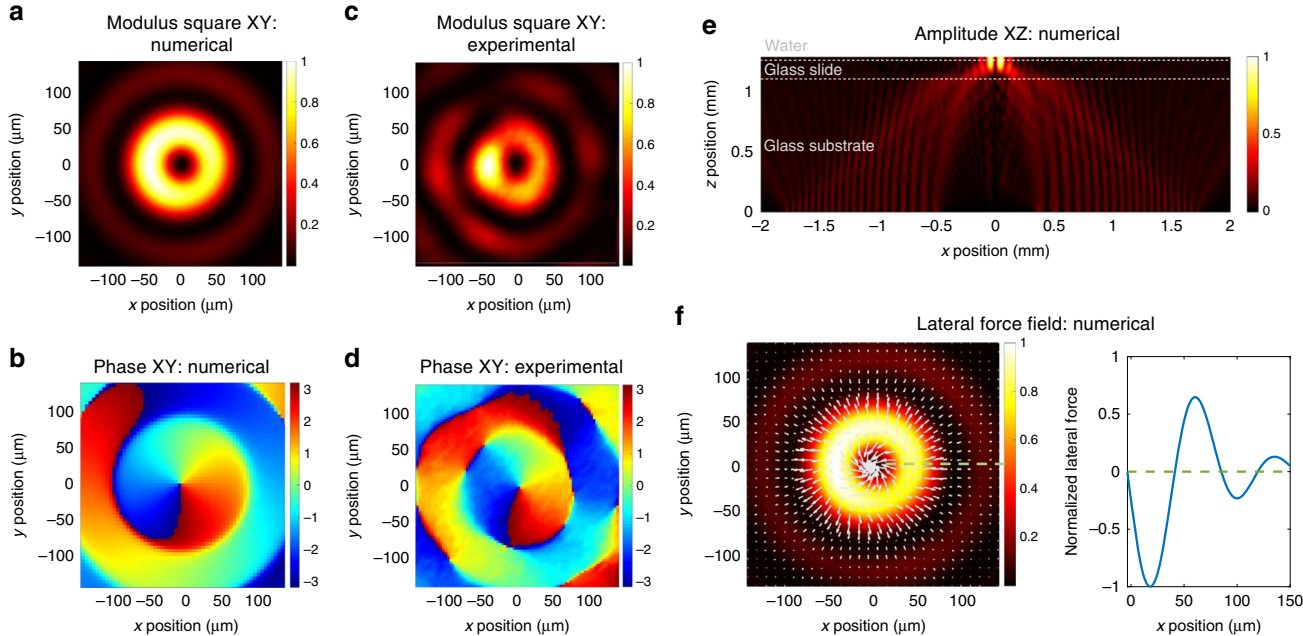

**Fig. 2 Acoustic field and radiation forces. a–d** Numerical predictions (**a**, **b**) and experimental measurements (**c**, **d**) with a UHF-120 Polytec laser Doppler vibrometer of the normalized modulus square (**a**, **c**) and phase (**b**, **d**) of the acoustic normal displacement at the surface of the glass slide (XY plane). The displacement magnitude is normalized by its maximum value measured to lie between 0.1 and 1 nm depending on the electrical power applied to the transducers. **e** Simulated evolution of the amplitude of the acoustic field in the propagation plane (XZ) from the source to the center of the channel. This simulation illustrates the concentration of the acoustic energy through focalization. **f** Normalized magnitude and distribution of acoustic forces. Left: the white arrows show the convergence of the force field toward the center of the beam but also that the first ring is repulsive for particles located outside the trap. Right: Magnitude of the lateral force along the green dashed line plotted in the left figure. When the force is negative, the particle is pushed toward the center of the acoustic vortex, while when it is positive it is pushed outward. Zero values correspond to static equilibrium positions. The magnitude of the maximum trapping force computed with the code varies between 30 and 650 pN (see "Methods" section "Simulations of the acoustic field and radiation force") for vibration amplitude of 1 nm (acoustic power of 2 mW) depending on the exact cells acoustic properties[42, 43].

force also show that as long as a cell is located at a distance of ≤40 μm from the center of the vortex, it is attracted toward the center of the beam (the lateral force is negative). This distance corresponds approximatively to the first ring radius and defines the spatial selectivity of the tweezers.

Finally, the temperature increase due to Joule heating in the electrodes as well as the total temperature increase due to both Joule heating and acoustic wave absorption was measured using an infrared camera to assess potential impact on biological material (see "Methods" section "Measurement of the thermal dissipation"). For most experiments presented in this paper (corresponding to acoustic displacement < 0.6 nm), the temperature increase is lower than 2.2 °C after 2 min of manipulation and even vanishes for the lowest power (0.1 nm). It reaches a maximum value of 5.4 °C at the top of the glass slide and 5.5 °C inside a drop of glycerol placed on top of the glass slide (acting as a perfectly absorbing medium) at the highest power used for high speed displacement of the cells. These measurements indicate that the first source of heat is Joule heating in the electrodes which could be solved by active cooling of the transducer. They also suggest that even at the largest power used in the present experiments, the moderate temperature increase remains compatible with cells' manipulation, as assessed in the next section. Indeed, the thermal increase, even in the worst-case scenario remains lower than the 6 °C recommended to ensure tissues' safety in medical imaging (thermal index of 6).

**Cells' manipulation, positioning and viability**. Cell manipulation is demonstrated in a microfluidic device integrated in a standard inverted microscope (Fig. 1e) to illustrate the fact that

our approach can be easily transposed to standard microbiology experiments. The device is composed of a thin glass slide treated to prevent cell adhesion and a PDMS chamber of controlled height (38 μm). The cells are loaded by placing a drop of the cell suspension (10–20 μL) on the glass surface using a micro-pipette and carefully lowering the chamber on top of the drop. The position of the vortex core is spotted with four triangular marks deposited at the surface of the glass substrate. Using an XY positioning system it is thereafter possible to align the tweezers' center to any cell present in the chamber. Upon activation of the AC driving signal, a cell situated inside the vortex core is nearly instantaneously trapped.

The first demonstration of the selective nature of our tweezers is showcased by our ability to pick up a single cell (breast cancer cell MDA-MB-231, 7 ± 1 μm in radius) among a collection of cells and move it along a slalom course where other free-to-move cells act as poles (see Fig. 3a, Supplementary Movie 2). Then a second cell initially serving as a slalom marker is moved to prove that it was free (Supplementary Movie 2). The precise displacement can be performed in any direction as demonstrated by the square motion of a cell around another (Fig. 3b, Supplementary Movie 3). Displacement can be performed even in the presence of other cells without any risk of coalescence as the first ring acts as a barrier (see Supplementary Movie 4, part 1). As can be seen in Fig. 2c, the radius of the first repulsive ring is typically 40 μm. This repulsive ring can also be used to separate a single cell from a cluster by activating the tweezers with the repulsive barrier located between the target cell and the other cells. In this way the target cell is attracted toward the vortex center while the other is expelled (see Supplementary Movie 4, part 2). Note also that the lateral force reaches a maximum for a distance ~20 μm from the

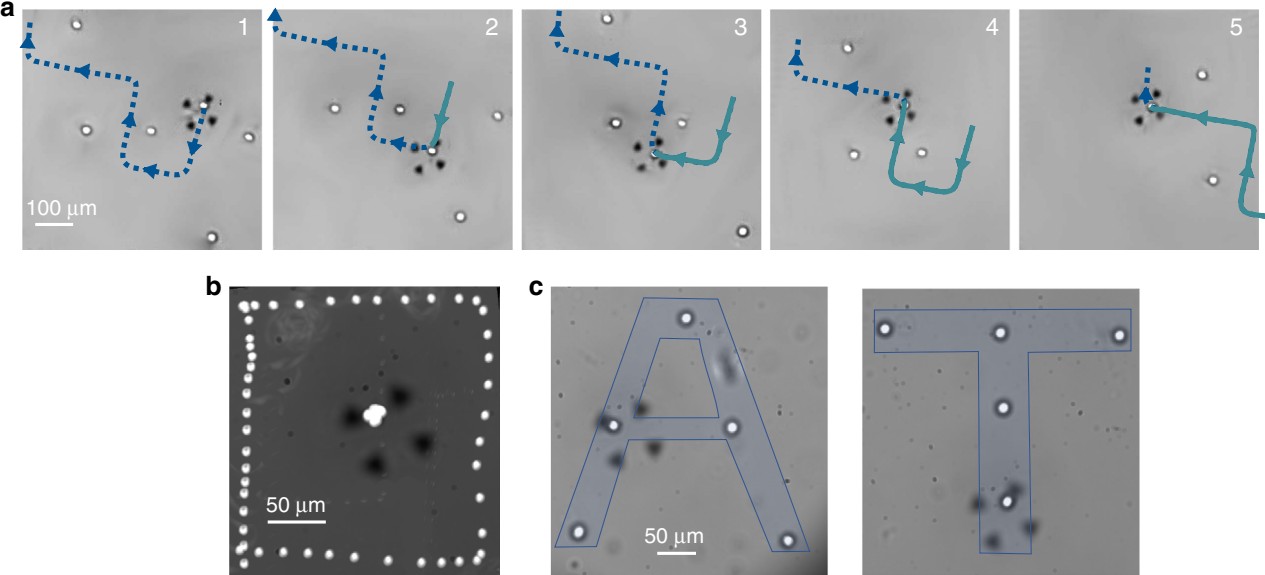

**Fig. 3 Cells' manipulation with selective acoustical tweezers. a** Stack of images illustrating the selective manipulation of a human breast cancer cell (MDA-MB-231) of radius 7 ± 1 μm between other cells. The blue dotted line and green continuous line show, respectively, the future and past path followed by the cell (see also Supplementary Movie 2). **b** Image illustrating the square relative motion of a trapped cell 1 of 7 ± 1 μm (located in the center of the picture) around another cell 2 obtained by superimposing the images of the two cells in the frame of reference of the trapped cell (see also Supplementary Movie 3). In this frame of reference, the successive positions of cell 2 form a square. For the sake of clarity other cells appearing in the field of view have been removed. **c** Manipulation of ten MDA cells (average radius 9 μm to form the letters A and T of the words Acoustical Tweezers). This alignment procedure was reproduced twice. Note that in these pictures the focus is voluntarily left under-focused to improve contrast of the cells.

center and then decreases until it reaches 0 at 40 μm. Because of this, a cell can be moved closer than 40 μm from another cell if there is a slight adherence of the cells on the substrate (see Supplementary Movie 5). Adhered cells can then be detached by increasing the acoustic power. Note also that second ring of much weaker intensity can also slightly affect free cells at large power.

One of the key ability enabled by selective acoustical tweezers is the capture, positioning, and release of cells at precise locations. As an illustration, a total of ten individual MDA cells were therefore positioned to spell the letters A and T of the words Acoustical Tweezers (Fig. 3c). The total manipulation time to achieve these results was kept under 10 min (<2 min per cell). All the operations represented in Fig. 3 were performed with acoustic vibration displacements <0.5 nm.

Finally, we performed some experiments to quantify the forces that can be exerted on cells with these tweezers. For this purpose, cells were trapped and then moved with an increasing speed until it was ejected from the trap. Velocities up to 1.2 mm s$^{-1}$ before ejection have been measured for cell displacement of diameter 12 ± 1 μm trapped with an acoustic field of magnitude 0.9 nm in a microchamber of height 38 μm (see Supplementary Movie 6). This corresponds to a trapping force of 194 ± 35 pN according to Faxen's formula[45], which lies in the range predicted by theory (see "Methods" section "Simulations of the acoustic field and radiation force"). As a comparison, this force is one order of magnitude larger than the maximum forces (20 pN) reported by Keloth et al.[32] with optical tweezers and obtained with one order of magnitude less power (1.8 mW here compared to the 26.8 mW used for optical trapping). Note that even at these comparatively large trapping force, the mechanical index in the present experiments (≤0.15) remains far below the safety threshold (1.9) defined to ensure tissue safety for medical imaging. Furthermore, unlike with optical tweezers, it is still possible to substantially increase this force with acoustical tweezers by increasing the actuation power and improving the thermal management of the device, as most of the dissipated power

comes from the transducer and not from the direct absorption by the medium.

As described in the introduction, one of the main gains that can be expected from transitioning from optical to acoustical tweezers is the absence of deleterious effects of the latter when manipulating living cells. The short- and long-term viability was investigated using a fluorescent viability assay as well as post exposure cell observation. A first set of experiments was thus conducted to address the short-term viability of MDA cells on eight cells. The cells were captured for 2 min in the vortex at maximum power (amplitude 0.9 nm) to mimic a standard positioning sequence and observed for any sign of damage during manipulation and for 30 min afterwards. During manipulation, no increase of fluorescence was observed suggesting that the sound field does not induce membrane permeabilization which correlates with viability decrease[46]. After the tweezers were switched off, the cell did not display any increase of fluorescence and remained at an intensity well under the dead cells found nearby (5× to 10× lower). This supports that short-term damages produced by the acoustical tweezers are minimal.

It is however known that damages experienced by a cell can lead to its death for hours afterwards[47]. To assess the long-term impact of cell manipulation using acoustical tweezers, we performed a viability assay overnight. The MDA cells were seeded at 60 (%) confluence ratio in two glass devices with no surface treatment and left to re-adhere for 5 h. Nine cells located at different positions in the two different microfluidic chambers were exposed to the tweezers of acoustic vortex at maximum power for 2 min each. An observation of the cells was performed after 19 h (half the population doubling rate of MDA cells[48]) to compare their viability with a control region of the device (see Fig. 4a). No extra mortality was observed in the illuminated region (dead/live cell ratio of 3%) compared to the statistics performed on the overall device (dead/live cell ratio of 5%). This likely indicates that the dead cells are depositing randomly and that the tweezers do not provoke extra mortality. We also studied

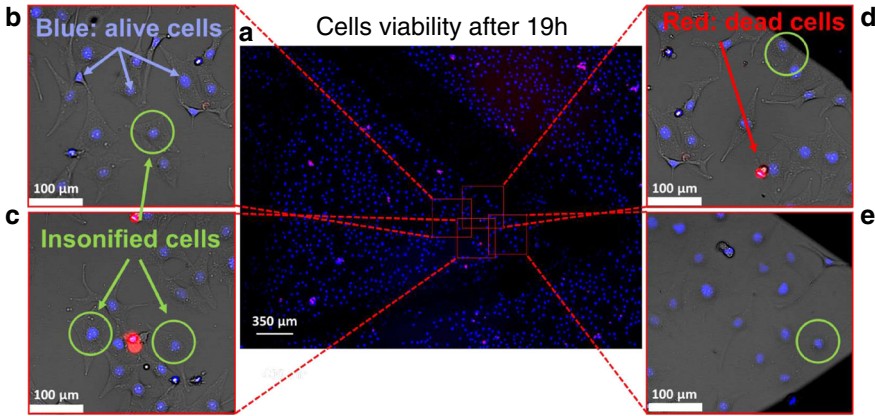

**Fig. 4 Cells' viability monitoring. a** Overview of the central part of the microfluidic device in which the viability experiments were performed. The cells are stained using a viability kit and imaged at 360 and 535 nm excitation (460 and 617 nm emission). The cell nucleus are represented in blue, while the dead cells appear in red. The whole field of view contains 4581 cells (226 dead—5%) while the region where manipulation took place contains 166 cells (5 dead —3%). **b–e** Details of the five cells exposed to the acoustical tweezers for 2 min (four others were exposed on another similar device). The green circle represents the first ring of the trap. Long-term viability tests were performed overall on nine insonified cells in two different microfluidic chambers.

in detail the fate of the nine illuminated individual cells (see Fig. 4b–e). All the cells exposed to the acoustic field (the green circle indicates the extension of the first ring of the vortex) and their immediate neighbors were alive and showed no difference compared to the nearby cells. In addition to short- and long-term cells' viability assays, it would be interesting in future work to investigate manipulated cells division with time lapse on longer observation time to ensure that cells are not forced into senescence.

## Discussion

In this work, cell selective manipulation is demonstrated through the capture and precise positioning of individual cells among a collection in a standard microscopy environment. Both short- and long-term viability of manipulated cells is evaluated, showing no impact on cells' integrity. This opens widespread perspectives for biological applications wherein precise organization of cells or microorganisms is a requisite. In addition, trapping force over wave intensity ratio two orders of magnitude larger than the one obtained with optical tweezers is reported with no deleterious effect such as phototoxicity. In future work, both the trapping force and selectivity could be further improved for 2D manip- ulation by increasing the tweezers' working frequency (see Sup- plementary Note). Based on considerations on the dissipation of acoustic waves in water, one can indeed envision acoustical tweezers working up to several hundred MHz. The applied force could be also increased by improving thermal management of the device to limit Joule heating. In this way, it would be possible to apply stresses several orders of magnitude larger than with optical tweezers without altering cells' viability, a promising path for acoustic spectroscopy[12], cell adhesion[49] or cell mechano- transduction[50–52] investigation. Indeed, the calibration of these tweezers would enable to apply controlled stresses to cells and monitor their response in force ranges not accessible before with other contactless tweezers. Furthermore, additional abilities could be progressively added to these tweezers: The focused vortex structure used for selective particle trapping in this paper is also known to exhibit 3D trapping capabilities[17,39]. This function was not investigated here owing to the confined nature of the microchamber but could closely follow this work. Synchronized vortices could also be used to assemble multiple particles, as recently suggested by Gong and Baudoin[53]. This would enable the investigation of tissue engineering[54] and envision 3D cell print- ing. Finally, the most thrilling and challenging perspective to this

work might be the future development of Spatial Ultrasound Modulators (analogs to Spatial Light Modulator in optics), designed to manipulate and assemble many objects simulta- neously. While such a revolution is on the way for large particles' manipulation in air[55–57], it would constitute a major break- through at the microscopic scale in liquids wherein the actuation frequencies are three orders of magnitude larger. The present work constitutes a cornerstone towards widespread applications of acoustical tweezers for biological applications.

## Methods

**Tweezers' design.** The tweezers were designed by materializing, with metallic electrodes, the hologram of a spherically focused acoustical vortex at the surface of an active piezoelectric substrate following Baudoin et al.[30]. The hologram was discretized on two levels resulting in two intertwined spiraling electrodes of inverse polarity, whose polar equations (electrodes centerline) are given by:

$$\rho_1 = \frac{1}{k}\sqrt{(\theta + C_2)^2 - (kz)^2}, \tag{1}$$

$$\rho_2 = \frac{1}{k}\sqrt{(\theta + C_2 + \pi)^2 - (kz)^2} \tag{2}$$

with ($\rho$, $\theta$, $z$) the cylindrical coordinates (the subscripts 1 and 2 stand for the electrodes 1 and 2), $k = \omega/c_a$ the wavenumber, $\omega = 2\pi f$ the angular frequency, $f$ the driving frequency of the system, $c_a$ the sound speed in the glass. The width of the electrodes equally distributed on both sides of the centerline defined by Eqs. (1) and (2) is kept equal to half the distance between the two electrodes. In the experiments described in this paper, three slightly different transducers were used: The first transducer (referred as Tweezer 1) was designed to excite transverse wave (sound speed $\approx$ 3500 m s$^{-1}$) in a D263T borosilicate glass substrate of thickness 1.1 mm (provider: PGO Online). The electrodes have an inner radius of 0.75 mm and outer radius of 1.6 mm, hence describing eight turns around the central axis. Since the speed of sound in glass substrates can vary substantially depending on the exact fabrication process, the precise resonance frequency was determined with a laser Doppler vibrometer (see "Methods" section "Microfluidic device fabrication"). The measured value was 47 MHz, used as the actuation frequency. These tweezers were used for the experiments reported in Fig. 3a (Supplementary Movie 2), Supple- mentary Movies 4 and 5, the second part of Supplementary Movie 6 and the viability experiments shown in Fig. 4. Note that a transverse wave in the solid can produce a longitudinal wave in the fluid as long as the incidence is not normal. Indeed, while longitudinal and transverse modes are two independent modes in the bulk of a solid they are coupled at an interface. The second transducer (Tweezer 2) was used to produce longitudinal wave (sound speed $\approx$ 3200 m s$^{-1}$) in a glass SF 57 HT ULTRA (provider: Schott). The resonance frequency determined with the vibrometer was 43.5 MHz. The electrodes had an inner radius of 0.25 mm and outer radius of 2 mm, hence describing 12 turns around the central axis. This transducer was used in its longitudinal excitation mode to perform experiments reported in Fig. 3b, c and Supplementary Movie 3. The third and last transducer was similar to the second transducer but with only ten turns of the electrodes. This transducer was used for the determination of the speed of displacement (Supple- mentary Movie 6, first part) and for the comparison of the acoustic field repre- sented in Fig. 2c, d. The advantage of the second and third transducers is that the

smaller inner radius of the electrodes results in weaker secondary rings. The advantage of the first transducer is that the type of glass matches with the glass slide (coverslip also in borosilicate) resulting in better transmission of the acoustic signal from the glass substrate to the microfluidic chamber and thus higher intensities at same actuation power.

**Tweezers' fabrication**. The active material used for the deposition of the electrodes is a 0.5-mm-thick, 3 inches diameter Y-36 cut Lithium Niobate piezoelectric substrate (LiNbO$_3$). The fabrication process starts with the deposition of the spiral metallic electrodes. First, the substrate is cleaned for 3 min in an ultrasound bath with acetone and isopropylic alcohol and dried with nitrogen (N$_2$). Then, promoter adherence (HMDS) and AZnLOF2020 resist is spread on the substrate with a thickness of about 2.9 μm. An optical mask and MA/BA6 SUSS Microtec UV Optical aligner are used to transfer patterns into the resist. After development, a period of 10 s oxygen plasma is performed in a Reactive Ion Etching system. Then, the substrate is slightly etched with Ar+ plasma at 150 eV for 90 s before to be metallized in situ with Ti/Au (40/400 nm) by evaporation. Lift-off is then realized in SVC-14 remover at 70 °C. At the end, the substrate is sonicated at 35 kHz at 15% power to enhance the lift-off operation. Substrate is cleaned in IsoPropylic Alcohol and dried in a nitrogen flow. A glass substrate of borosilicate D263 T (PGO Online) or SF 57 HT ULTRA (Schott) with a diameter of 56.8 and 65 mm respectively and a thickness of 1.1 mm is then glued with an optically transparent epoxy glue (EPOTEK 301-2) at the surface of the piezoelectric substrate. The gluing process is critical to ensure good transmission of the wave from the piezoelectric to the glass substrate and avoid losses in the glue. Here the substrates were glued with a layer of ~1 μm of epoxy glue obtained by (i) cleaning properly the piezoelectric and glass substrates with acetone, isopropylic alcohol and dicholoromethane to improve glue spreading, (ii) depositing with a pipette a controlled volume of glue (calculated to form a uniform layer of 1 μm after spreading) at the center of the piezoelectric substrate, (iii) positioning the glass substrate on top of the piezoelectric substrate, (iv) leaving the glue spread by capillarity in a vacuum chamber with a control horizontality until it covers the whole surface between the Niobate and the glass substrates. Note that the glue was degassed prior to use to avoid the formation of bubbles. Since the coefficient of transmission of energy from epoxy to glass is around 64%. This means that most of the acoustic energy is transmitted after four round trips in the glue. Since 8 × 1 μm remains small compared to the attenuation length in epoxy, our gluing process is expected to ensure good transmission from the Niobate to the glass. A 15-nm-thick chromium layer is then deposited by evaporation on the upper face without etching that will serve for the markers. A vacuum box is used to remove bubbles formation after the mixing step. The curing step takes 2 days at room temperature on a marble horizontal plan. The alignment is manually realized with coarse cross to place it in the center of the LiNbO$_3$ substrate to facilitate the next photo-lithography step. A cleaning step is realized with acetone, isopropylic alcohol and nitrogen drying for only few seconds to remove dust. A backside alignment photo-lithography is performed with a AZ1505 resist layer of thickness about 0.5 μm with the same optical alignment system as before to report the cross target in the center of the transducers on the glass surface. After development, the glass substrate is placed for 10 s in oxygen plasma in RIE system to remove resist traces. The Cr layer is etched in Cr etching solution. At last, cleaning step is used to remove resist on the glass.

**Microfluidic device fabrication**. The fluidic device in which the cells are manipulated is made of a thin (150 μm) glass substrate glued to a Plexiglas (PMMA) adapter frame to make it compatible with our positioning system. The adapter frame was made out of a 4-mm-thick PMMA sheet which was cut using TROTEC LASER system. The glass surface was coated with Cytop to prevent excessive adhesion of cells on the surface during manipulation. The Cytop coating solution was prepared by diluting a CTL-809M polymer solution with CT-solv 180 solvent (1 ml:10 ml) (v/v). A few drops of this solution were deposited by spin coating (1500 rpm 300 m s$^{-2}$ for 30 s) on the top of a cleaned (acetone and IPA and dried using nitrogen gas) glass slide (22 mm × 50 mm coverslip). The thickness of the Cytop layer was measured to be around 30 nm. A curing step was performed in an oven at 180 °C for 30 min. The glass slide was glued to the bottom of the PMMA adapter frame using NOA61 glue and dried by exposing it to UV light for 1 min. To fabricate the PDMS chamber, an Si wafer was cleaned using Piranha solution (mixing H$_2$SO$_4$ sulfuric acid with H$_2$O$_2$ hydrogen peroxide = 5:1 (v/v)) for 10–15 min. Negative SU8-2035 photoresist was spin coated on the Si wafer using the following protocol: 800 rpm, ramp 1000 m s$^{-2}$ for 10 s and 2250 rpm, ramp 1000 m s$^{-2}$ for 30 s resulting in a resist thickness of 45 μm. The resist is soft baked on a hot plate at 65 °C for 3 min and 95 °C for 6 min.

The mask prepared for the chamber's design was inverted on the Si wafer with a hard contact type exposed to UV light (365 nm) for 20 s with a power of 10 mW cm$^{-2}$. After the exposure, a hard bake takes place on a hot plate at 65 °C for 2 min and 95 °C for 6 min. Then, the Si wafer was developed using SU8 developer for around 5 min, rinsed with IPA and further dried with N$_2$.

PDMS (40 g base: 4 g curing agent) 10/1 (w/w) was mixed and placed in a vacuum box to remove all the air bubbles from the mixture, for about 15 min. The PDMS was poured on the Si wafer that has been prepared previously. It was cured in the oven at 110 °C for 10 min. Finally, the PDMS was cut, peeled and cleaned

**Table 1 Lateral trapping force as a function of cells' compressibility and density.**

| Cell density (kg m$^{-3}$) | Cell compressibility (×10$^{-10}$ Pa$^{-1}$) | Lateral force (pN) |
|---|---|---|
| 1000 | 3.3 | 650 |
| 1000 | 4.4 | 30 |
| 1210 | 3.3 | 290 |
| 1210 | 4.4 | 450 |

Here the results are given for an actuation frequency of 43.5 MHz, an acoustic displacement of 1 nm and cell radius of 10 μm.

with IPA and dried with N$_2$. After placing the cells on the top of the glass slide, the PDMS slice is put on the top of the cells' solution.

**Experimental characterization of the acoustic field**. The acoustic field out-of-plane normal displacement was measured at the top of a metallized glass slide (to improve reflectivity) with a Polytech UHF-120 laser Doppler vibrometer equipped with a Mitutoyo M plan Apo ×20 objective. The measures were performed in FFT mode by applying a chirp signal (typically ranging from 10 to 50 MHz), which enables a frequency treatment of the data. For the determination of the amplitude of vibration, the tweezers were excited with the exactly same frequency generator (IFR 2023 A) and amplifier (AR50A250 Amplifier 150 W) as the one used in the experiments.

**Simulations of the acoustic field and radiation force**. The acoustic field was calculated with an angular spectrum Matlab code which consists of (i) the 2D Fourier Transform of the signal in a source plane (which turns the signal into a sum of plane waves), (ii) the propagation of these plane waves to the target plane through the different layers of materials and (iii) the inverse Fourier transform of the signal. The source plane is obtained from the exact shape of the transducer. For the simulations of the acoustic field presented in Fig. 2, we used the following parameters corresponding to the third transducer: glass substrate:thickness 1.1 mm, sound speed 3200 m s$^{-1}$/glass slide: thickness 150 μm, sound speed 5300 m s$^{-1}$/water: thickness 40 μm, sound speed 1481 m s$^{-1}$. The actuation frequency was 43.5 MHz. Note that we did not consider in this code the evolution of the transmission coefficients depending on the incidence angle. The force field was calculated with the analytical formula provided by Sapozhnikov and Bailey[44], using the numerically computed acoustic field. To determine the magnitude of the force, we used the vibration magnitude measured experimentally, i.e. 1 nm of displacement. We chose a typical cell size of 10 μm in radius, closely matching the average size measured experimentally with a cell counter. Since the acoustic properties (density and compressibility) of the MDA cells were not known, we performed some calculations of the force for different properties corresponding to the extreme values of the density and compressibility reported in the literature[42,43]. The resulting maximum trapping force is summarized in Table 1.

The properties of the growth medium was approximated to be the ones of water. We can note that the force varies significantly depending on the cells' properties. This dependence is used in acoustophoretic systems to sort cells depending on their properties.

**Experimental estimation of the trapping force**. To estimate the trapping force, a cell of diameter $D = 12 \pm 1$ μm was trapped with transducer 1 and moved with the $XY$ stage inside a microfluidic chamber of height $H = 38$ μm at an increasing speed $U$ until the cell was ejected from the trap. Speed up to $U_{max} = 1.2$ m s$^{-1}$ have been measured. The trapping force $F_f$ was then calculated with Faxen's formula derived for a particle moving between two infinite walls:

$$F_f = 3\pi\mu D U \times \cdots$$
$$\left[1 - 0.04(D/H) + 0.418(D/H)^3 + 0.21(D/H)^4 - 0.169(D/H)^5\right]^{-1},$$

with $\mu = 0.001$ Pa s the dynamic viscosity, leading to a force of 194.5 ± 35 pN (here we suppose that the particle is located at the center of the channel and is rigid). Nevertheless, since Faxen's formula is only valid for low aspect ratio $D/H$ (here $D/H \approx 0.3$), we further performed some direct numerical simulation of the force exerted on the particle with the so-called SIMPLE Navier−Stokes solver implemented in OpenFoam code. With this code, no significant departure from the analytical formula is found (see Fig. 5) since we also obtain a force of ≈194.5 pN.

**Microscope observation**. All cells' manipulation were performed under observation using an inverted microscope (IX71 model, Olympus Corp, Japan) equipped with a ×20 ApoPlan objective (Olympus Corp, Japan) and either a Raptor camera (Raptor Photonics, USA) at 7 fps or a Photometrics camera sCMOS Back Illuminated Prime-BSI at 40 fps. Images were post-treated with ImageJ software. To enable easy positioning and manipulation of the cells, the microfluidic device is

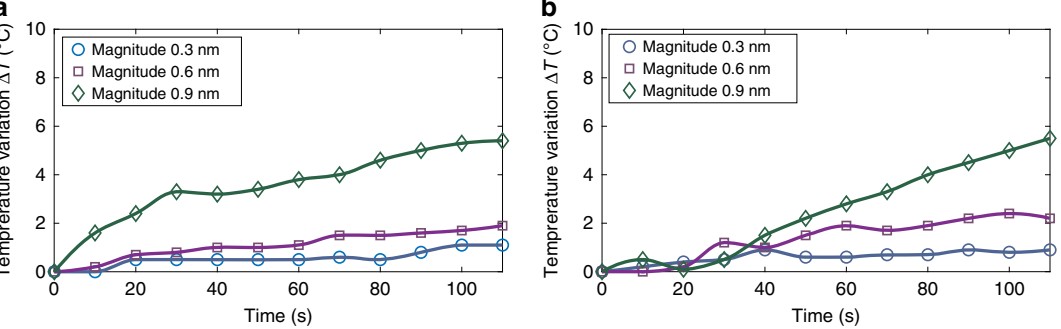

**Fig. 5 Simulation of the drag force exerted on a rigid particle moving in a microchannel.** Here simulations are performed for particles of diameter 12 μm moving in a channel of height 38 μm. **a** Velocity field in two orthogonal planes intersecting the particle. **b** Pressure field in the central plane parallel to the walls. **c** Convergence of the force depending on the grid refinement level.

**Fig. 6 Temperature variation induced by acoustical tweezers.** The measurements were performed with an infrared camera Testo 871 for different acoustic displacement magnitude. **a** Measurement at the top of the glass slide. **b** Measurement inside a glycerol layer deposited on top of the glass slide and acting as a perfectly absorbing layer. The data represented on these graphs represent single measurements. These measurements were conducted twice with similar results.

positioned on top of the tweezers with a drop of silicon oil to ensure good conduction of the acoustic waves. A Plexiglas chip holder design was produced using a TROTEC LASER system so that the holding frame of the microfluidic device fits perfectly inside it. Displacement is performed using a *XY* stage (Thorlabs PLS-XY), which provides for adaptable speed parameters and precise control in the two axes.

**Cell culture**. The chosen cell line, MDA-MB-231, is cultured in Dulbecco's modified Eagle medium (DMEM, Eurobio) supplemented with 10% Fetal Bovine Serum(FBS) (Qualified, US origin, Standard sterile filtered, Gibco), 0.4% Penicillin-Streptomycin (10,000 U ml$^{-1}$, Gibco) and 1% L-Glutamine (Gibco, 200 mM). Cells are detached from the cell culture flask using 0.05% trypsin-ethylenediaminetetraacetic(EDTA) (Gibco) and suspended in DMEM medium. The cell suspension is characterized using a life science cell counter (Countess II FL Automated Cell Counter, Invitrogen, USA) which measured their radius at 9 ± 3 μm and the cell density at $2 \times 10^6$ cells ml$^{-1}$ before being used inside the device.

**Cells viability monitoring**. *Short-term validity:* In order to monitor short-term cell viability, the MDA-MB-231 cells are marked after trypsinization and dilution using the fluorescent cell viability kit (ReadyProbes cell viability imaging kit, blue/red, Invitrogen, USA) at one drop per ml of cell suspension. Then the cells are loaded on the glass substrate previously treated with Cytop, covered with the PDMS chamber and manipulated using the procedure described in the main text at the maximum acoustic field magnitude (0.9 nm) for 2 min. The 360 and 535 nm excitation and the 460 and 617 nm emission, corresponding respectively to live and dead cells, are imaged regularly in an Olympus IX71 microscope equipped with fluorescence filters, both during the manipulation and up to 30 min afterward to monitor their evolution.

*Long-term validity:* In this case, after trypsinization, the cells are marked using the fluorescent viability kit (ReadyProbes cell viability imaging kit, blue/red, Invitrogen, USA) and then loaded on the nontreated glass slide using a micropipette. Some lines (visible in Fig. 4a) were drawn with a black marker on the lower face of the glass substrate to improve cells' localization. Thereafter, the substrate is placed in an incubator at 37 °C temperature and 5% $CO_2$ level for 120 min to enable cells to adhere on the surface and ease their subsequent long-term localization. A few cells located on different parts of the substrate are insonified for 2 min at the maximum power used for cells' manipulation. After this sequence, the substrate is placed again in an incubator at 37 °C temperature and 5% $CO_2$ for 19 h. After this time, fluorescence imaging of the substrate is performed in the Leica

DMi8 microscope in both areas where manipulations have taken place as well as nonexposed areas to monitor long-term viability.

**Estimation of the acoustic power**. The average acoustic power $P_a$ can be calculated from the acoustic intensity vector $\mathbf{I} = p_a \mathbf{v}_a$ (with $p_a$ and $\mathbf{v}_a$ the acoustic pressure and velocity perturbations, respectively) according to the following formula:

$$P_a = \left\langle \iint_S \mathbf{I} \cdot \mathbf{n} \, dS \right\rangle_t$$

with $\langle f(t) \rangle_t = 1/T \int_0^T f(t) dt$ the time averaging operator, $t$ the time, $T$ the period, $S$ a surface intersecting the beam and $n$ the normal vector. With the vibrometer we measured the normal acoustic displacement $d_a^n$ to the surface of the glass slide, which follows a spherical Bessel beam distribution and can be approximated in this plane by a field of the form:

$$d_a^n = \frac{d_{max}}{j_1^{max}} j_1(kr) e^{i(\omega t - \varphi)}$$

with $(r, \varphi)$ the polar coordinates, $d_{max}$ the maximum displacement measured with the vibrometer in the section, $j_1^{max}$ the maximum of the spherical Bessel function of first order, $j_1$ the spherical Bessel function of first order, $\omega$ the angular frequency and $k$ the wavenumber characterizing the lateral field variation at the top of the glass slide. Indeed, it has been shown that the lateral field variation is well approximated by a spherical Bessel function[30]. The pressure and normal velocity fields can then be estimated according to: $v_a^n = \mathbf{v}_a \cdot \mathbf{n} = \omega d_a^n$ and $p_a \sim \rho_o c_o v_a^n$, with $\rho_o$ the fluid density (note that this latter expression is exact only for plane waves and thus constitutes an estimation of the pressure level in the present case). We thus obtain:

$$P_a = \frac{\rho_o c_o \omega^2}{2} \left[ \frac{d_{max}}{j_1^{max}} \right]^2 2\pi \int_{r=0}^{\infty} j_1(kr)^2 dr.$$

The normal displacement used for cells' manipulation varied typically between 0.1 and 1 nm giving an acoustic power varying between 20 μW and 2 mW.

**Measurement of the thermal dissipation**. The temperature increase induced by the tweezers was monitored with an infrared camera (Testo 871) to assess the impact on biological material. The temperature increase was measured every 10 s at

different power (corresponding to vibration magnitudes ranging from 0.1 to 0.9 nm). The temperature was measured first at the top of a glass slide metallized with titanium and then in a drop of glycerol deposited on top of the glass slide acting as a perfectly absorbing medium for acoustic waves (owing to its extremely large viscosity). The first measurement gives an indication of the temperature increase due to Joule heating in the electrodes, while the second gives an indication of the total temperature increase due to both Joule heating and acoustic wave absorption. At the lowest power (0.1 nm displacement), no significant temperature variation is reported. The results for the magnitudes 0.3, 0.6 and 0.9 nm are represented in Fig. 6. For intensities lower than 0.6 nm, the temperature variation does not exceed 2.2 °C while at the highest power (displacement of 0.9 nm) an increase of temperature of 5.4 °C is measured at the top of the glass and 5.5 °C inside the drop of glycerol after 2 min. Note that the camera has been calibrated with a hot plate to determine the emissivity of titanium and glycerol.

## Data availability

The data that support the findings of this study are available from the corresponding authors upon reasonable request.

## Code availability

The angular spectrum Matlab code used for the numerical simulations of the acoustic vortex propagation and the trapping forces estimation presented in Fig. 2 is available in Supplementary Information.

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

## Acknowledgements
This work was supported by ISITE-ULNE (Prematuration and ERC Generator programs), SATT Nord, Institut Universitaire de France and Renatech Network. We acknowledge the platform WaveSurf from Université Polytechnique Hauts de France for performing the measurements with the laser Doppler vibrometer.

## Author contributions
M.B., J.-L.T. and A.V. designed the research; M.B., J.-C.G, R.A.S. and A.V. built the setup and performed the experiments; A.S. performed cells culturing and characterization; M.B. and Z.G. performed the acoustic numerical simulations; M.B. and P.F. performed the hydrodynamic numerical simulations; M.B., O.B.M., R.A.S. and N.S. performed the acoustic field characterization with the laser Doppler vibrometer; M.B., O.B.M., J.-L.T. and A.V. analyzed and interpreted the results; M.B., J.-L.T. and A.V. wrote the paper. All authors approved the final version of the manuscript.

## Competing interests
The authors declare no competing interests.
