## [Peer Review File · Nature Communications]

Reviewers' Comments:

Reviewer #1:

Remarks to the Author:

This paper presents an important step beyond that in [8] in which the manipulation of inanimate 150um particles was shown with a 4MHz vortex device. The same device design principles are used here, but in the present paper a 45MHz device is shown. The extension is non-trivial, requiring enhancements of most detailed aspects of the design. Beyond the device manufacture difficulties, the present device is important as it gives access to selective cell manipulation. So, whilst the device in [8] was scientifically interesting, the present device is practically interesting. Indeed, although the present device was very much predictable from [8], it is nonetheless exhilarating to see it working so well – the videos make this particularly clear. The forces applied to cells and acoustic power are nicely analysed and quantified. Also, the heating effect is measured and found to be only a few degrees. These steps to characterise the device show it to be a potential competitor for optical tweezers, again, making this an important step. Initial cell viability has been assessed and looks promising, although cell proliferation post sonication was not shown. In summary, this is an important piece of work and should be rapidly published.

Prior to publication, the following point should be considered.

Line 38 – the power levels in medical imaging are much lower than here, so don't think this is relevant. Hence, this discussion should be re-worded, or some precise quantification used.

Line 43 – the attenuation of ultrasound varies strongly with frequency. It does not seem reasonable to say that its uniformly "weak" across this whole frequency range. Indeed, this statement raises the obvious question as to what is the upper operating frequency of this type of device? This would be interesting to explore in the discussion.

Line 51 – the authors need to comment on the microbeam tweezers of Shung at USC, e.g. doi: 10.1002/bit.23073. These operate in the ray-acoustic regime. I suggest the relative merits of the microbeam approach vs. the vortex are also mentioned briefly in the discussion.

Line 188 – clarify that the field is sinusoidal in form and the quoted displacements are the amplitude of this wave.

Line 193 – can the authors state the energy efficiency of the device, i.e. how much electrical power was input to generate this acoustic power?

Line 258 – what is the closest approach of the manipulated cell with respect to another static cell – this defines the spatial selectivity. Can two touching cell be separated?

Line 284 – why is this difference (between optical and acoustic tweezers) only one order of magnitude. If one looks at the difference in velocity of sound relative to light, one would expect a much larger improvement (5 orders of magnitude). Presumably the difference is due to the shape of the field used here relative to that used in optical tweezing, i.e. the acoustic field extends around and away from the cell, in the present device, whereas in optical trapping the energy is focused on the particle.

Line 335 – would be better to say, "apparently harmless", as a full viability study is not in place yet. But I agree that the results are extremely encouraging.

Overall/discussion – can the authors comment on the range of cell-sizes and inanimate object sizes trapped with the present device. This would help readers understand the capabilities. If this is not known, it would also be worth stating this.

Reviewer #2:

Remarks to the Author:

The authors report upon an ultrasonic methods for a "precise selective contactless manipulation and positioning of human cells in a standard microscopy environment, without altering their viability."

Major issues:

1. The paper uses terms like – “cell selective manipulation”, “precise and selective” and “without altering their viability”, which to a biologist have specific meanings. I don't believe the paper evidences these claims. The authors must provide detailed quantitative information showing precise, cell selective separation. Presently, the paper only shows simple 2D manipulation of small numbers of cells.

The term “without altering viability” can have many meanings and at the least needs controlled data, although really data is needed showing that mechanosensitive receptors are not activated (see later).

2. The authors have already published all of the different technical abilities described in this publication in previous papers (e.g. tweezing and wavefront shaping). The novelty here is therefore low technically, the authors have simply increased the frequency with little engineering challenge. In truth they just show 2D patterning of cells which has been demonstrated acoustically and optically, or electro-optically.

The merit in publishing this work will only arise through the context of the technique's performance in enabling a clear analytical application for quantitative cell screening (point 1 above), which currently could not be achieved using other acoustic, flow or optical techniques. Using the technique to place a few cells is incremental.

Technical issues (all must be addressed in detail)

1. Epoxy is a viscoelastic material and will heat up due to the acoustic wave propagation. I am not convinced that most of the heat is generated is due to the Joule heating. When epoxy is used to stick a glass slide to an ultrasonic transducer, the polymer will heat up considerably, in particular at the frequencies and powers used.

2. On the subject of acoustic frequency, data needs to be shown quantitatively as why the frequency chosen was used. For example, why not 60 MHz or 80 MHz?

3. The authors do not consider possible resonance frequencies of cells and related biophysical processes; for example, the response of suspended chondrocytes over a range of frequencies using continuous ultrasound has a primary resonant frequency of 5.2 ± 0.8 MHz (<https://doi.org/10.1371/journal.pone.0181717>). The author's statement is not corroborated or evidenced.

Cells are highly variable depending upon their history and their current environment. There is no evidence that the cells used were in anyway “controlled” in their state (biological, cycle, etc). Mechanical stimulation through the acoustic forces, even in the lifetime of the experiment will alter the cell biochemistry and biophysics. This needs discussion.

Specific Comments (all must be answered)

1. Abstract: "2 mW of driving power" - is this the acoustic power?

2. Page 1, Column 2, Lines 41-45:

The Frequency range is very subjective - it can go up to 1 GHz or more;

3. Unsupported claim: "the weak attenuation of sound in both water and tissues at these frequencies limits absorption induced thermal heating"; This needs to be clearly referenced or demonstrated experimentally in ESI.

4. Page 3, Column 1, Lines 134-135: delete "remote". It's not remote.

5. Page 3, Column 1, Lines 159-161: unsupported claim "poor thermal conductivity of glass". Relative to what, Water? Air? I think not.

6. Page 7, Column 1, Line 399: "slow sound speed" - it is a subjective claim. Again, as above. Water?, Air?
7. Page 7, Column 1, Line 383: "The first transducer was designed to excite transverse wave" - how does the transverse wave couples to fluid with cells?
8. Page 7, Column 1, Line 397: Why mention a "transverse wave sound speed" transducer, if it is not used in this study?
9. Page 8, Column 1, Lines 518-521: authors state: "For the determination of the amplitude of vibration, the tweezers were excited with the exactly same frequency generator (IFR 2023 A) and amplifier (AR Amplifier 50 W) as the one used in the experiments." - what was the output power from the generator to the amplifier?
10. Fig. 6: Amplitude must be replaced by magnitude in both parts of the illustration.

Reviewer #3:

Remarks to the Author:

This well-written manuscript describes a new kind of acoustical tweezer designed to capture single cells. The main experimental novelty is the miniaturisation of a 4.4 MHz acoustic tweezer to 47 MHz frequency, which insures a much smaller precision. An acoustical vortex generates a trap whose size is such that a maximum force occurs 20 micrometers away from the trap center, and then the tweezer force vanishes.

Experiments confirm the ability to move single cells among other ones that remain untouched. The manuscript is containing extensive confirmation of the efficiency of the trap, with measurements of the actual sound field, temperature increase measurements, drag force measurements and even cell viability tests. Numerical simulations show the trapping ability and the spatial profile of the sound field.

Overall, this study is opening new realms for acoustic manipulation with increased precision, and brings physical evidence on the mechanisms at play. I suggest publishing the manuscript provided corrections are incorporated to address the following issues on the description of the tweezer:

1. Line 31. « But, the dramatically lower speed of sound compared to light leads to driving power several orders of magnitude smaller. ». The term « dramatically» sounds strange here, with a negative side, that does not fit the positive side.
2. Line 131. Figures are not called in order in the text, which creates confusion, since the reader has to search information in an apparent random order. Please fix this issue in other places of the manuscript, by changing the text or the figure order.
3. Figure 2C: what is exactly the scale for the measured displacement?
4. Figure 2E. Focalisation occurs at a z position that seems higher than the 1.1mm of the experiments. Please explain the implications of this difference.
5. Figure 2F. Is the numerical prediction for the trapping force in agreement with the experiments? The manuscript is not really clear on this aspect.
6. There is a non negligible pushing force in between 50 and 70 micrometers. Did the author observe such a repulsion force when dragging a cell nearby another that is very close?
7. What would happen if the density of cells would be very important, with distances between cells less than 20 micrometers?
8. Not much is said on the axial acoustic force. In principle it could push away the cell from the substrate, or attract the cell, depending on the actual position of the focus of the transducer. Please comment on this aspect.

Modification de l'article :

Response to reviewer #1

This paper presents an important step beyond that in [8] in which the manipulation of inanimate 150um particles was shown with a 4MHz vortex device. The same device design principles are used here, but in the present paper a 45MHz device is shown. The extension is non-trivial, requiring enhancements of most detailed aspects of the design. Beyond the device manufacture difficulties, the present device is important as it gives access to selective cell manipulation. So, whilst the device in [8] was scientifically interesting, the present device is practically interesting. Indeed, although the present device was very much predictable from [8], it is nonetheless exhilarating to see it working so well – the videos make this particularly clear. The forces applied to cells and acoustic power are nicely analysed and quantified. Also, the heating effect is measured and found to be only a few degrees. These steps to characterise the device show it to be a potential competitor for optical tweezers, again, making this an important step. Initial cell viability has been assessed and looks promising, although cell proliferation post sonication was not shown. In summary, this is an important piece of work and should be rapidly published.

We thank the referee for underlining the importance of this work and the need for its rapid dissemination, and above all for his/her constructive advice, which helped us improve the quality of the manuscript.

Prior to publication, the following point should be considered.

Line 38 – the power levels in medical imaging are much lower than here, so don't think this is relevant. Hence, this discussion should be re-worded, or some precise quantification used.

The reviewer is right: this point needs to be clarified. In fact, the power used is far below the recommended levels for medical imaging. Hence no damage to cells is expected from these criteria. The accepted power levels in medical imaging are ruled by two indexes:

- *First, the mechanical index, which should remain below 1.9 is designed to avoid inertial cavitation. This index is the negative peak acoustical pressure in MPa divided by the square root of the frequency in MegaHertz. For instance, at 45 MHz the pressure should remain below $1.9 \sqrt{45} = 14$ MPa. Here the maximum pressure magnitude of the first ring remains below 1 MPa (for the maximum acoustic power reported in this experiment), which corresponds to a mechanical index of 0.15 well below the recommended threshold in medical imaging.*
- *Second, the thermal index, which is the ratio between the relevant attenuated acoustic power at the depth of interest and the estimated power necessary to raise the tissue equilibrium temperature by one degree Celcius. The FDA recommends a thermal index smaller than 6, i.e. a temperature increase lower than 6°C. In the present experiments (i) the temperature increase is always under 6°C as attested by our temperature measurements, and (ii) this temperature increase is mostly due to heating induced by the transducer, not by absorption of the wave by the medium. Indeed, the attenuation length in water at 45 MHz is of the order of 2 cm, so several orders of magnitude larger than the depth of the channel. So, absorption induced heating will be weak leading to a very low thermal index (following its rigorous definition).*

We modified the paragraph in the introduction as follows:

“In addition, the innocuity of ultrasounds on cells and tissues below cavitation and deleterious heating thresholds defined by the mechanical and thermal indexes is largely documented^{34–38} and demonstrated daily by their widespread use in medical imaging³⁹. Indeed, the frequencies typically used in medical ultrasound (1 MHz to 100MHz) and in the present work (~ 45MHz) are far below electronic or molecular excitation resonances thus avoiding adverse effects on cells integrity. In addition, the attenuation in water at these frequencies (attenuation lengths between 4m and 4mm) remains weak for manipulation at the micrometric scale, hence limiting absorption induced thermal heating. Note that the mechanical index in the present experiments ≤ 0.15 remains far below the safety threshold (1.9) defined for medical imaging.^{37,38} Finally, almost any type of particles (solid particles, biological tissues, drops) can be trapped without pre-tagging³¹ and the low speed of sound enables spatial resolutions down to micrometric scales even at these comparatively low frequencies. “

and added the following sentence at the end of section III:

“Indeed, the thermal increase, even in the worst-case scenario remains lower than the 6°C recommended to ensure tissues safety in medical imaging (Thermal index of 6).”

Line 43 – the attenuation of ultrasound varies strongly with frequency. It does not seem reasonable to say that its uniformly “weak” across this whole frequency range. Indeed, this statement raises the obvious question as to what is the upper operating frequency of this type of device? This would be interesting to explore in the discussion.

This is a good point, that we indeed need to discuss: For classic fluids the attenuation scales quadratically with frequency and linearly in biological tissues. But in this last case a lot of the effect is due to scattering by inhomogeneities. For cells in-vitro manipulation with acoustical tweezers, the attenuation medium is hence mainly water. In water, we can estimate the evolution of the attenuation length L_a of ultrasound as a function of the driving frequency: 40m at 1 MHz, 2 cm at 45 MHz, 4 mm at 100 MHz. The central question then becomes: to which characteristic dimension L_c should one compare the acoustic attenuation length? The wave propagates in the fluid over the height of the channel (typically 2-5 times the diameter of the cells for 2D manipulation). The dimension of the trapped object being typically equal (or inferior to) the wavelength (see discussion below) we can take the characteristic propagation length as 5 times the wavelength. If we impose that only 10% of the injected power is absorbed over this length (to avoid too much heating), this leads to a criterion: $L_a / L_c > 50$. From this criterion we obtain a limit frequency of about 500 MHz for the manipulation of objects sensitive to thermal heating.

This is now discussed in the introduction (see previous answer), in the conclusion:

“In future work, both the trapping force and selectivity could be further improved for 2D manipulation by increasing the tweezers working frequency (see Method section K). Based on considerations on the dissipation of acoustic waves in water, one can indeed envision acoustical tweezers working up to several hundred MHz.”

and a complete new section in Methods (Section K: Discussion on the possibility and limits of vortex-based tweezers in terms of particle size, selectivity and driving frequency)

Line 51 – the authors need to comment on the microbeam tweezers of Shung at USC, e.g. doi: 10.1002/bit.23073. These operate in the ray-acoustic regime. I suggest the relative merits of the microbeam approach vs. the vortex are also mentioned briefly in the discussion.

The differences between the two approaches are examined below:

With the focused beam (FB) approach operating in the Mie regime, particles trapping at the center of the beam (focal point) is only possible (i) if the rays can penetrate the particles (i.e. for weak and specific acoustic contrasts), (ii) when the object is significantly larger than the wavelength and (iii) at specific frequencies (depending on the resonances of the object). Otherwise, the object would be expelled from the focal point. With acoustical vortices (AV) however, trapping is possible for particles with positive contrast factors smaller than the dimension of the first ring (see Fig. 1 below), and at the center of the vortex (constituting a silent zone).

Figure 1: Lateral force F_x exerted on a cell as a function of the cell position x , for different cell radii a for the acoustic beam represented on Fig. 2A, an acoustic maximal displacement (magnitude) of 1nm and cells density and compressibility of 1100 kg m^{-3} and $4 \times 10^{-10} \text{ Pa}^{-1}$ respectively. This figure shows that the lateral force is negative and hence ensure a stable trap only for particles whose radius is lower than 45 microns. This corresponds approximately to the lateral extension of the first ring (maximum amplitude) of 38 microns. The maximum trapping force is obtained for $a = 35$ microns.

The advantages of the vortex beams approach are that: (i) Any particle with positive acoustic contrast factor (i.e. cells, rigid particles, liquids denser than the surrounding medium) can be trapped at the center of the vortex. (ii) A large span of particle size can be trapped at a given frequency (basically all particles smaller than the size of the first ring ($\sim 0.7 \lambda$, with λ the wavelength) for which the radiation pressure dominates over acoustic streaming. (iii) The particle lies in the silent zone of the beam. Hence stresses are applied on the trapped object only if an external force (liquid drag, ...) moves the particle away from the beam center. (iv) Lower actuation frequencies lead to lower dissipation. Conversely working in the Mie Regime enables stronger gradients of the acoustic field.

In addition:

(i) 3D trapping capabilities have never been demonstrated with focused beam while they have been demonstrated with vortex beams [Baresch et al., Phys. Rev. Lett. 116: 024301 (2015)].

(ii) The device proposed in the present paper and in our previous Science Adv. paper is transparent, flat, miniaturized and hence can be easily integrated in a classic microscopy environment, which is not the case for the other devices proposed in the literature.

This discussion has been summarized as follows in section II of the paper:

“Compared to tweezers based on focused beam operating in the Mie regime, the vortex-based tweezers enable to trap objects with positive contrast factors at the beam center, in 3D, and at lower operating frequencies, hence limiting deleterious heating. Conversely these lower frequencies (and hence wavelength) lead to weaker gradients compared to tweezers operating in the Mie regime.”

Line 188 – clarify that the field is sinusoidal in form and the quoted displacements are the amplitude of this wave.

Modified as suggested:

“The final device hence consists of (see Movie 1 in SI, Fig. 1A, 1C, 1E): (i) spiraling holographic transducers excited with a sinusoidal electrical signal”... “The magnitude of the sinusoidal acoustic field”

Line 193 – can the authors state the energy efficiency of the device, i.e. how much electrical power was input to generate this acoustic power?

To obtain a good electrical efficiency, many features of the system must be optimized. In particular the impedance of the transducer must be matched to 50 Ohms to avoid spurious reflection of the electric energy back to the source. Since in our case, we were not limited by the electrical power of the source (since our amplifier could deliver up to 150W), we did not optimize at all the electrical aspects of the device. This is why we decided to indicate the acoustic power instead of the electrical power (which does not reflect the capabilities of the technology here). Unfortunately, due to the coronavirus outbreak we don't have access to the lab to measure the electrical power actually used for the excitation of the device. The best we can say with the information that we have at our disposal (frequency generator power levels and amplifier amplification levels) and the datasheet of our amplifier (AR 150A250) is that if the transducer was adapted to 50 Ohm (which is not), the electrical power would lie between 1.58mW and 1.58W at the power used in the experiments.

Line 258 – what is the closest approach of the manipulated cell with respect to another static cell – this defines the spatial selectivity. Can two touching cells be separated?

The trap dimension is visible on Fig. 2F of the paper. The lateral force remains attractive up to a distance of $\sim 40\mu\text{m}$ from the beam center, which corresponds to the lateral radius of the first ring. Any free cell whose center is located in a radius of $40\mu\text{m}$ will feel an attractive force. This defines the spatial selectivity. Since this spatial selectivity is directly related to the radius of the first ring (whose spatial extension is $\sim 0.7\lambda$), it can be improved (for 2D trapping) by increasing the excitation frequency, with an optimal spatial selectivity obtained when the radius of the ring is equal to the radius of the trapped object (which also corresponds to the maximum force). Indeed, our simulations (Fig. 1 above) shows that it is possible to trap objects (in 2D) as large as the radius of the first ring (while 3D trapping requires smaller particles [Baresch et al., Phys. Rev. Lett. 116: 024301 (2016)]).

Nevertheless,

(i) Even if the spatial selectivity is larger than the distance between two cells, it is still possible to separate them by activating the tweezers with the repulsive barrier (first ring) located between the two cells. In this case, one cell is attracted toward the vortex center, while the other is expelled. This strategy was used to collect a cell from a cluster of free cells and then move it to another cluster. We added a new video that shows this manipulation now referred as movie M4.

(ii) Two cells can be approached to a smaller distance than these 40 microns if there is a weak adherence of one of the cells to the substrate. Then, the acoustic power can be increased to unpin an adhered cell and move it in turn. Since all the experiments presented in the first version of the paper were performed with surface treatments to avoid cell adherence, we added a video to illustrate this point (now referred as Movie M5).

The spatial selectivity and the two additional movies are now discussed in the manuscript:

Section III: “These simulations of the lateral force also show that as long as a cell is located at a distance $\leq 40\mu\text{m}$ from the center of the vortex, it is attracted toward the center of the beam (the lateral force is negative). This distance corresponds approximatively to the first ring radius and defines the spatial selectivity of the tweezers. “

Section IV: “This repulsive ring can also be used to separate a single cell from a cluster by activating the tweezers with the repulsive barrier located between the target cell and the other cells. In this way the target cell is attracted toward the vortex center while the other is expelled (see Movie 4, part 2). Note also that the lateral force reaches a maximum for a distance $\sim 20\mu\text{m}$ from the center and then decreases until it reaches zero at $40\mu\text{m}$. Because of this a cell can be moved closer than $40\mu\text{m}$ from another cell, if there is a slight adherence of the cells on the substrate (see movie M5). Adhered cells can then be detached by increasing the acoustic power.”

and perspectives on how improving the selectivity are provided in the “Conclusion and outlook section”:

“In future work, both the trapping force and selectivity could be further improved for 2D manipulation by increasing the tweezers working frequency (see Method section K). Based on considerations on the dissipation of acoustic waves in water, one can indeed envision acoustical tweezers working up to several hundred MHz.”

Line 284 – why is this difference (between optical and acoustic tweezers) only one order of magnitude. If one looks at the difference in velocity of sound relative to light, one would expect a much larger improvement (5 orders of magnitude). Presumably the difference is due to the shape of the field used here relative to that used in optical tweezing, i.e. the acoustic field extends around and away from the cell, in the present device, whereas in optical trapping the energy is focused on the particle.

You are right that the energy is focused on a ring of one wavelength rather than a spot of half this size. This account for a factor of 4 for the force. Note that this decrease is mitigated by are reduced exposure for the cell since it is located in a zone of silence. But the difference between the wavelength is much more important. In optics the wavelength is around $1\mu\text{m}$ and is here 30 times larger. So, in optics with the same input energy, the energy density is 3 orders of magnitude larger at the focus. The gradient scale with the wavelength so again more force is expected in optics. So, we would say that the main reason is the difference in wavelength. As discussed in some of the previous answers and shown on Fig. 1, we could increase the frequency significantly and indeed gain several orders of magnitudes on the force. Here we chose to use some frequencies that are compatible with 3D trapping capabilities (by upscaling the systems presented in ref. [Baresch et al., Phys. Rev. Lett. 116: 024301 (2016)]). Indeed, if the radius of the ring becomes comparable with the radius of the particle, the radial trap is increased considerably, but the axial trap is lost. Since we envision to test 3D cells trapping in the future, we designed our tweezers with this constraint in mind.

This question is now discussed in details in the Method section K, and in the previous text already introduced to answer the previous questions.

Line 335 – would be better to say, “apparently harmless”, as a full viability study is not in place yet. But I agree that the results are extremely encouraging. Overall/discussion.

We performed both short term and long-term cell viability assays which shows no impact on cells viability. The acoustic pressure is also well below the thresholds (mechanical and thermal index) defined for medical imaging, as discussed previously. Finally, there have been many studies on the impact of (non-selective) tweezers on cells at comparable power that are referenced in our introduction. Nevertheless, as underlined by the reviewer, cells mechanics is extremely complex and

further investigation is necessary to assess precisely all the impact of the manipulation on cells biology, that may also differ depending on the type of cells considered.

So as suggested we qualified our claim and removed harmless from the conclusion since the viability tests performed in the manuscript are detailed in the next sentence:

“Both short-term and long-term viability of manipulated cells is evaluated, showing no impact on cells integrity.”

and the mechanical and thermal indexes are discussed in the Introduction.

Can the authors comment on the range of cell-sizes and inanimate object sizes trapped with the present device. This would help readers understand the capabilities. If this is not known, it would also be worth stating this.

In view of the questions raised by the reviewer, we performed numerical simulations of the cell-size that can be trapped with the present tweezers. This shows (as we expected) that cells of radius up to the radius of the first ring can be trapped and that the lateral force increases when the radius of the cell increases. As discussed previously, we chose this frequency since we envision now to investigate 3D trapping with these tweezers and that axial trap is only possible when the trapped object is significantly smaller than the first ring size [Baresch et al., Phys. Rev. Lett. 116: 024301 (2016)]. Indeed, the object needs to lie in the silent zone to avoid too much axial force coming from the progressive part of the wave.

The new Method section K discusses this question in details.

Response to reviewer #2

The authors report upon an ultrasonic methods for a “precise selective contactless manipulation and positioning of human cells in a standard microscopy environment, without altering their viability.”

Major issues:

1. The paper uses terms like – “cell selective manipulation”, “precise and selective” and “without altering their viability”, which to a biologist have specific meanings. I don’t believe the paper evidences these claims. The authors must provide detailed quantitative information showing precise, cell selective separation. Presently, the paper only shows simple 2D manipulation of small numbers of cells. The term “without altering viability” can have many meanings and at the least needs controlled data, although really data is needed showing that mechanosensitive receptors are not activated (see later).

The reviewer is right, terms may have different meanings depending on the community. We now clearly define these terms in the manuscript:

In the abstract we now refer to “spatial selectivity”.

In the introduction we define these terms:

“Here “selectivity” refers to spatial selectivity, i.e. the ability to select and move an object independently of other neighboring objects.”

“Cells viability was assessed following exposure to the acoustic field measured by short-term and long-term fluorescence viability assays.”

In addition,

(1) As suggested by the reviewer we further added a video of manipulation in a concentrated suspension of free cells, showing (i) the possibility to pick up a single cell among a cluster and bring it to another cluster (Movie M4, second part) and (ii) the possibility to move it in between other free cells (movie M4, first part). We also now discuss and show a movie (movie M5) illustrating a case wherein there is a slight adherence of the cells on the substrate. In this case it is possible to navigate very closely to the other particles and then increase the power to unpin a previously stuck cell.

These videos are now discussed in section IV:

“As can be seen in Fig. 2C, the radius of the first repulsive ring is typically 40 μm . This repulsive ring can also be used to separate a single cell from a cluster by activating the tweezers with the repulsive barrier located between the target cell and the other cells. In this way the target cell is attracted toward the vortex center while the other is expelled (see Movie 4, part 2). Note also that the lateral force reaches a maximum for a distance $\sim 20 \mu\text{m}$ from the center and then decreases until it reaches zero at 40 μm . Because of this, a cell can be moved closer than 40 μm from another cell if there is a slight adherence of the cells on the substrate (see movie M5). Adhered cells can then be detached by increasing the acoustic power.”

The possibility to pick-up and move a single cell independently of other neighboring particles was already clearly evidenced by our movies M2 and M3.

(2) We also added a discussion in the introduction on the effect of ultrasounds on tissue and cells and in particular compared the mechanical and thermal indexes in our experiments to the norms defined to ensure ultrasound medical imaging safety (especially of insonified fetuses) by ultrasounds:

“Indeed, the frequencies typically used in medical ultrasound (1 MHz to 100MHz) and in the present work ($\sim 45\text{MHz}$) are far below electronic or molecular excitation resonances thus avoiding adverse effects on cells integrity. In addition, the attenuation in water at these frequencies (attenuation lengths between 4 m and 4 mm) remains weak for manipulation at the micrometric scale, hence limiting absorption induced thermal heating. Note that the mechanical index in the present experiments ≤ 0.15 remains far below the safety threshold (1.9) defined for medical imaging.”

“Indeed, the thermal increase, even in the worst scenario remains lower than the 6°C recommended to ensure tissues safety in medical imaging (Thermal index of 6).”

Finally, we must underline that these questions of cells viability have been investigated by other authors with standing-wave based acoustical tweezers, in the references mentioned in the introduction section (ref [34-37]) at acoustic powers comparable to the one described in this paper.

2. The authors have already published all of the different technical abilities described in this publication in previous papers (e.g. tweezing and wavefront shaping). The novelty here is therefore low technically, the authors have simply increased the frequency with little engineering challenge.

Manipulating 150 microns inert rigid particles and 12 microns living cells are two completely different challenges and we believe that any team that has been confronted to such challenge would attest this point. It took more than one year of the work of a dedicated team to switch from one study to the other. Manipulating living objects requires to ensure their viability. Ensuring this viability

while (i) increasing the frequency by an order of magnitude and (ii) dramatically increasing the required acoustic power to apply significant forces on objects with low acoustic contrasts is far from trivial. Indeed, many dissipation effects scale as the frequency or the frequency square. In addition, cells are objects with extremely low contrast factors compared to the surrounding fluids. Since the acoustic radiation force depends on the density and compressibility contrast factors with the surrounding medium, manipulating cells is far more complicated than manipulating solid particles, and requires higher intensity to apply the same force. Thus limiting thermal heating required to reconsider many aspects of the wave synthesis system, such as the transducers designs to ensure optimal focusing and reduced dissipation by optimizing the number of spirals, their width, the thickness of the metallization, adding radial electrodes, ... but also paying attention to gluing aspects which become fundamental (as discussed later on following one of the reviewer question). But ensuring cells safety while enabling their manipulation, also required to define a whole biological protocol with specific microfluidic chamber designs, surface treatments to avoid their adherence, ... many questions that were not at all considered in our previous work.

This paper also presents several physics related advances: We investigate an aspect that is (i) fundamental to compare our technology to others, (ii) not evoked in our previous paper and (iii) far from “a little engineering challenge”: predicting and measuring the force applied on the cells. Treating this question required (i) to build on a full numerical code able to compute the force applied on a cell from the real acoustic field radiated from our tweezers in a regime wherein the Long Wavelength Approximation does not hold, (ii) characterize precisely the radiated acoustic field with a laser interferometer, and (iii) design experiments to measure this force experimentally and compare to our numerical predictions.

In truth they just show 2D patterning of cells which has been demonstrated acoustically and optically, or electro-optically. The merit in publishing this work will only arise through the context of the technique's performance in enabling a clear analytical application for quantitative cell screening (point 1 above), which currently could not be achieved using other acoustic, flow or optical techniques. Using the technique to place a few cells is incremental.

Cells selective displacement (as defined above) with acoustical tweezers has never been demonstrated. This is why this is the central message underlined in both the title of our paper and the abstract. So, the novelty does not lie in 2D cells patterning that could indeed be achieved with other techniques (often with significant drawbacks in their uses), but in the showcased ability to select a single cell, move it independently of neighboring particles and position it. Hence with our technique, we could select specific cells (e.g. from different cell lines) and organize them spatially, which is not possible with standing-wave based tweezers. We also demonstrate forces of up to 200pN with low power ultrasounds. This combined selectivity, forces range and biocompatibility has not been achieved with any other device. The limits of optical devices for life science are thoroughly discussed in the introduction.

Technical issues (all must be addressed in detail)

1. Epoxy is a viscoelastic material and will heat up due to the acoustic wave propagation. I am not convinced that most of the heat is generated is due to the Joule heating. When epoxy is used to stick a glass slide to an ultrasonic transducer, the polymer will heat up considerably, in particular at the frequencies and powers used.

The reviewer is perfectly right. This is indeed one of the crucial issues that we had to solve to carry out this work. To evaluate the acoustic energy absorbed in the epoxy layer, one must compare the acoustic attenuation length with the distance of propagation in this layer. For a layered medium the distance of propagation depends on how many back and forth trip a progressive wave has to do before

being fully transmitted to the next layer. The coefficient of transmission of energy from epoxy to glass $\frac{4Z_1Z_2}{(Z_1+Z_2)^2} = 16/25$ (64%) is rather large since the impedance ratio is around 4. This means that roughly 4 round trips are expected before the energy is fully transmitted. To avoid too much dissipation in the glue, the epoxy layer thickness was reduced to a thickness of 1 micron. This was obtained by (i) cleaning properly the glass and niobate wafer prior to glue deposition (to obtain surface of high energy improving liquid spreading by capillarity), (ii) depositing with a pipette a controlled volume of glue calculated to form this layer of 1 micron after spreading and (iii) leaving the glue spread by capillarity in a vacuum chamber until it covers the whole surface between the glass slide and the niobate lithium. Note that the glue was degassed prior to use to suppress any bubble. So, 4 round trips in a layer of 1 micron leads to 8 microns, which is small compared to the wavelength in the epoxy glue (around 40 μm at 45 Mhz) which itself is small compared to the attenuation length in epoxy glues. Due to this optimization of the gluing process, we are confident that it is not responsible for the temperature increase. We now discussed thoroughly this point in the Method section A:

“The gluing process is critical to ensure good transmission of the wave from the piezoelectric to the glass substrate and avoid losses in the glue. Here the substrates were glued with a layer of $\sim 1\mu\text{m}$ of epoxy glue obtained by (i) cleaning properly the piezoelectric and glass substrates with Acetone, Iso-Propylic Alcohol and Dichloromethane to improve glue spreading, (ii) depositing with a pipette a controlled volume of glue (calculated to form a uniform layer of 1 micron after spreading) at the center of the piezoelectric substrate, (iii) positioning the glass substrate on top of the piezoelectric substrate, (iv) leaving the glue spread by capillarity in a vacuum chamber with a control horizontality until it covers the whole surface between the Niobate and the glass substrates. Note that the glue was degassed prior to use to avoid the formation of bubbles. Since the coefficient of transmission of energy from epoxy to glass is around 64%. This means that most of the acoustic energy is transmitted after 4 round trips in the glue. Since $8 \times 1 \mu\text{m}$ remains small compared to the attenuation length in epoxy, our gluing process is expected to ensure good transmission from the Niobate to the glass.”

2. On the subject of acoustic frequency, data needs to be shown quantitatively as why the frequency chosen was used. For example, why not 60 MHz or 80 MHz?

This choice of the frequency was guided by ref. [Baresch et al., Phys. Rev. Lett. 116: 024301 (2016)]. Indeed, we envision, as a next step, to investigate 3D cells manipulation and hence designed our tweezers with our constraint in mind. Axial trapping with acoustical vortices is only possible when the trapped particle is significantly smaller than the radius of the first ring, i.e. when the particle mostly lies in the silent zone of the acoustical vortex. Otherwise, the axial force resulting from the progressive part of the one-sided vortex tends to push the particle away from the center and cannot be compensated by gradient forces. According to calculation performed in ref. [Baresch et al., Phys. Rev. Lett. 116: 024301 (2016)], a good compromise is achieved when the ratio $\frac{d}{\lambda} \approx 0.3$, which served as a guideline for the design of these tweezers.

Figure 2: Lateral force F_x exerted on a cell as a function of the cell position x , for different cell radii a for the acoustical beam represented on Fig. 2A, an acoustic maximal displacement (magnitude) of 1nm and cells density and compressibility of 1100 kg m^{-3} and $4 \times 10^{-10} \text{ Pa}^{-1}$ respectively. This figure shows that the lateral force is negative and hence ensure a stable trap only for particles whose radius is lower than 45 microns. This corresponds approximately to the lateral extension of the first ring (maximum amplitude) of 38 microns. The maximum trapping force is obtained for $a = 35$ microns.

On the other hand, as long as only 2D trapping is sought for, the precise working frequency of the tweezers is less critical since according to our calculation (see Figure 1 above), a large range of particle sizes can be trapped with a vortex at a given frequency. The trapping force and selectivity can be nevertheless further improved by increasing the tweezers working frequency so that the radius of the first ring ($\sim 0.35 \lambda$) matches with the radius of the particle. Indeed Figure 1 shows that the maximum force is obtained when the radius of the particle matches the radial extension of the first ring ($\sim 37 \mu\text{m}$), which also corresponds to the optimal selectivity since in this case, a trapped particle could be approach to another particle until they get in contact.

We now briefly discuss this point in the conclusion section:

“In future work, both the trapping force and selectivity could be further improved for 2D manipulation by increasing the tweezers working frequency (see Method section K). Based on considerations on the dissipation of acoustic waves in water, one can indeed envision acoustical tweezers working up to several hundred MHz.”

and thoroughly in a new dedicated section (Method Section K).

3. The authors do not consider possible resonance frequencies of cells and related biophysical processes; for example, the response of suspended chondrocytes over a range of frequencies using continuous ultrasound has a primary resonant frequency of $5.2 \pm 0.8 \text{ MHz}$ (<https://doi.org/10.1371/journal.pone.0181717>). The author’s statement is not corroborated or evidenced.

Interesting point since resonance may strongly affect the force applied on an object. Indeed, radiation pressure force is mainly dependent on the phase between incident and scattered wave and a phase shift occurs at the resonance. Nevertheless, most resonances appear for object larger than half the wavelength for pure geometric reasons. Some special cases like bubble in water are well known: bubbles much smaller than the wavelength (a few microns) have a resonant frequency in MHz range (mm wavelength). This feature is used by contrast agents. However, this is possible only with a large contrast of mass and compressibility between the bubble and the surrounding fluid: air and water. Indeed, a bubbly medium has roughly the density of water (since density of air can be neglected) and

the compressibility of air (since the bubble will compress first). In condensed matter (either liquid or solid), this cannot occur since the contrast are much weaker, especially for cells in water, which are mostly constituted of liquids. In echography numerous measurements have shown that soft biological tissues have an impedance comparable to water at 10 %. The speed of sound in tissue is roughly the same as in water.

Cells are highly variable depending upon their history and their current environment. There is no evidence that the cells used were in anyway “controlled” in their state (biological, cycle, etc). Mechanical stimulation through the acoustic forces, even in the lifetime of the experiment will alter the cell biochemistry and biophysics. This needs discussion.

This is indeed a very interesting point. Moving a cell by any technique (acoustic, optic, magnetic, flow,...) requires to apply mechanical stresses on the cells. Indeed, these mechanical stresses can affect the activity of some specific cells. This is why we think that acoustical tweezers is a promising tool to investigate mechanotransduction as long as the stresses applied on the cell can be calibrated. This is also one of the things we want to investigate in the future, in collaboration with biologists. This is now discussed in the conclusion section:

“In this way, the application of stresses several orders of magnitude larger than with optical tweezers without altering cells viability, a promising path for acoustic spectroscopy²⁹, cell adhesion⁵⁵ or cell mechano-transduction¹²⁻¹⁴ investigation. Indeed, the calibration of these tweezers would enable to apply controlled stresses to cells and monitor their response in force ranges not accessible before with other contactless tweezers. “

Specific Comments (all must be answered)

1. Abstract: "2 mW of driving power" - is this the acoustic power?

Indeed, we now make it clear that it is the acoustic power.

2. Page 1, Column 2, Lines 41-45: The Frequency range is very subjective - it can go up to 1 GHz or more; 3. Unsupported claim: "the weak attenuation of sound in both water and tissues at these frequencies limits absorption induced thermal heating"; This needs to be clearly referenced or demonstrated experimentally in ESI.

The reviewer is right, this statement is incomplete. We have modified this paragraph to clarify our thought. First, we now refer to frequencies typically used in medical ultrasound, i.e. frequencies between 1 and 100 MHz. Second, we removed “tissue” since for cells in-vitro manipulation with acoustical tweezers, the attenuation medium is mainly water. In water, waves attenuation scales as the square of the frequency. We can estimate the evolution of the attenuation length L_a of ultrasound as a function of the driving frequency: 40m at 1 MHz, 2 cm at 45 MHz, 4 mm at 100 MHz. The central question is then: to which characteristic dimension L_c comparing the acoustic attenuation length? The wave propagates in the fluid over the height of the channel (typically 2-5 times the diameter of the cells for 2D manipulation), the dimension of the trapped object being typically equal (or inferior to) the wavelength (see discussion below) we can take the characteristic propagation length as 5 times the wavelength. If we impose that only 10% of the injected power is absorbed over this length (to avoid too much heating), this leads to a criterion: $L_a / L_c > 50$. From this criterion we obtain a limit frequency of about 500 MHz, so well above the 1 MHz-100MHz range.

So, this paragraph has been rewritten as follows:

“Indeed, the frequencies typically used in medical ultrasound (1 MHz to 100MHz) and in the present work (~ 45MHz) are far below electronic or molecular excitation resonances thus avoiding adverse effects on cells integrity. In addition, the attenuation in water at these frequencies (attenuation lengths between 4 m and 4 mm) remains weak for manipulation at the micrometric scale, hence limiting absorption induced thermal heating.”

We also discuss these questions in the new Method section K.

4. Page 3, Column 1, Lines 134-135: delete "remote". It's not remote.

Done.

5. Page 3, Column 1, Lines 159-161: unsupported claim "poor thermal conductivity of glass". Relative to what, Water? Air? I think not.

Glass is a dielectric and hence has a small thermal conductivity compared to metals where the free electrons play an essential role. Without the glass slide, the liquid would be in direct contact with the gold electrodes, which are the heat sources. This would result in dramatically increases thermal transfers. The glass space helps spread the heat generated in the electrode over a wider area, while slowing down the heating due to the lower thermal conductivity compared to the metal electrodes.

Since the term “poor thermal conductivity of glass” was indeed not clear, we removed it as the sentence was clear without it.

6. Page 7, Column 1, Line 399: "slow sound speed" - it is a subjective claim. Again, as above. Water?, Air?

We compare with others usual glass like borosilicate, which have a longitudinal and transverse wave speed of 5 500 and 3000 respectively compared to 3200 and 1800 for the SF 57 HT ULTRA glass. We removed this terminology since the values of the glass sound speed are given in the text and hence this comparison is not necessary.

7. Page 7, Column 1, Line 383: "The first transducer was designed to excite transverse wave" - how does the transverse wave couples to fluid with cells?

Transverse and longitudinal wave are two independent modes in an infinite solid. At an interface they are coupled by the boundary conditions and partially converted in one another. For a perpendicular incidence, the coupling with a liquid would be weak since only longitudinal wave exist in a liquid. We use strong focusing and here the coupling from transverse wave in the solid to longitudinal wave in the liquid is strong. We added a comment on this point:

“Note that a transverse wave in the solid can produce a longitudinal wave in the fluid as long as the incidence is not normal. Indeed, while longitudinal and transverse modes are two independent modes in the bulk of a solid they are coupled at an interface.”

8. Page 7, Column 1, Line 397: Why mention a "transverse wave sound speed" transducer, if it is not used in this study?

The first transducer is based on transverse waves and used in the Fig. 3A (Movie 2), the viability experiments shown in Fig. 4, in movies 4 and 5 as well as in the second part of Movie 6.

The second transducer can be used in both longitudinal and transverse mode simply by changing the actuation frequency. We thought originally that it might be interesting for the reader to know that the transducer can be used both in a transverse mode and a longitudinal mode by simply switching the actuation frequency. We have now removed these details to avoid confusion for the reader, since for this transducer, only results with the longitudinal mode are indeed presented.

9. Page 8, Column 1, Lines 518-521: authors state: "For the determination of the amplitude of vibration, the tweezers were excited with the exactly same frequency generator (IFR 2023 A) and amplifier (AR Amplifier 50 W) as the one used in the experiments." - what was the output power from the generator to the amplifier?

The output power from the generator to the amplifier was varying between 0.1 and 1 mW.

10. Fig. 6: Amplitude must be replaced by magnitude in both parts of the illustration.

Done.

Response to reviewer #3

This well-written manuscript describes a new kind of acoustical tweezer designed to capture single cells. The main experimental novelty is the miniaturisation of a 4.4 MHz acoustic tweezer to 47 MHz frequency, which insures a much smaller precision. An acoustical vortex generates a trap whose size is such that a maximum force occurs 20 micrometers away from the trap center, and then the tweezer force vanishes. Experiments confirm the ability to move single cells among other ones that remain untouched. The manuscript is containing extensive confirmation of the efficiency of the trap, with measurements of the actual sound field, temperature increase measurements, drag force measurements and even cell viability tests. Numerical simulations show the trapping ability and the spatial profile of the sound field.

Overall, this study is opening new realms for acoustic manipulation with increased precision, and brings physical evidence on the mechanisms at play. I suggest publishing the manuscript provided corrections are incorporated to address the following issues on the description of the tweezer:

We thank the reviewer for underlining the perspectives opened by this work and the extensive physical and biological work done to carefully characterize these tweezers. We also thank the reviewer for his/her valued advice, which helped us improve the quality and clarity of the manuscript.

1. Line 31. « But, the dramatically lower speed of sound compared to light leads to driving power several orders of magnitude smaller. ». The term « dramatically» sounds strange here, with a negative side, that does not fit the positive side.

Thanks for underlining this point. We have replaced this term by “drastically”.

2. Line 131. Figures are not called in order in the text, which creates confusion, since the reader has to search information in an apparent random order. Please fix this issue in other places of the manuscript, by changing the text or the figure order.

We thank the reviewer for his/her advice. We checked and the figure are first called in the right order. But since subfigures are called at multiple locations, it might be the reason for this feeling of apparent randomness. The difficulty is that each figure must also have its coherence and we unfortunately did not find a simple way reorganize the figures to match more closely with their call in the text while keeping their coherence.

3. Figure 2C: what is exactly the scale for the measured displacement?

The displacement is normalized by the maximum measured displacement. Indeed, while the field profile measured with the interferometer is not modified by a change in the driving electric power, the maximum value directly depends on the actuation power. Since different powers have been used in different experiments, we preferred to represent the normalized field. We modified the legend to clarify this point:

“The displacement magnitude is normalized by its maximum value measured to lie between 0.1 nm and 1 nm depending on the electrical power applied to the transducers.”

4. Figure 2E. Focalisation occurs at a z position that seems higher than the 1.1mm of the experiments. Please explain the implications of this difference.

The reviewer raises a good point. It was not clear from Figure 2E, where the glass substrate / glass slide interface and the glass slide / water interfaces are located. In fact, Figure 2E only shows the propagation of the acoustic field up to the central part of the channel (the upper part is truncated). So, focalization indeed occurs at the center of the channel. It is just that the depth of field is cut by the interface at the focus and hence result in this unusual shape compared to usual focusing picture observed in homogeneous media. We added some information on the figure (see below) and in the legend to clarify this point:

“Simulated evolution of the amplitude of the acoustic field in the propagation plane (XZ) from the source to the center of the channel.”

5. Figure 2F. Is the numerical prediction for the trapping force in agreement with the experiments? The manuscript is not really clear on this aspect.

The reviewer is right, this point is not very clear in the main part of the manuscript. In fact, the exact acoustic properties of our cells are not known. So, we performed a parametric study of the force dependence over the cell density and compressibility with the range of values reported in the literature. This analysis shows that the force is very sensitive on the properties of the cells. This is why acoustic system are used for cells sorting. We confirm that the measured force value lies in the range of predicted value but the error bar is large.

We added a sentence to clarify this point:

“This order of magnitude agrees with the maximum force measured experimentally (~ 200 pN) for similar parameters (see section IV).”

6. There is a non-negligible pushing force in between 50 and 70 micrometers. Did the author observe such a repulsion force when dragging a cell nearby another that is very close? 7. What would happen if the density of cells would be very important, with distances between cells less than 20 micrometers?

Indeed, this repulsive force is observed experimentally. We added a movie (movie M4) showing (i) how this repulsive force prevent coalescence with other cells when a single cell is manipulated in a dense suspension of cells (ii) how this repulsive force can be used to separate a single cell from a cluster. This video is now commented in the text:

“This repulsive ring can also be used to separate a single cell from a cluster by activating the tweezers with the repulsive barrier located between the target cell and the other cells. In this way the target cell is attracted toward the vortex center while the other is expelled (see Movie 4, part 2).”

So, this repulsive barrier defines our tweezers selectivity. Nevertheless, two cells can be approached to a smaller distance than these 40 microns if there is a weak adherence of the cell on the substrate. Then, the acoustic power can be increased to unpin an adhered cell and move it in turn. Since all the experiments presented in the first version of the paper were performed with surface treatments to avoid cell adherence, we added a video to illustrate this point (now referred as Movie M5).

8. Not much is said on the axial acoustic force. In principle it could push away the cell from the substrate, or attract the cell, depending on the actual position of the focus of the transducer. Please comment on this aspect.

Measuring the axial force created by our tweezers would indeed be a very interesting perspective to this work. The present experiments were designed to obtain an “as much as possible” 2D configuration by designing channels with a small depth (40 microns) compared to the lateral dimensions (millimetric). This configuration is highly desirable for microbiology experiments wherein it is necessary to have cells localized in a plane to be able to capture them easily and visualize them with a microscope with a small depth of field. Since in this configuration the axial force would be extremely perturbed by the walls, we are designing experiments in chambers with larger depth and means to visualize lateral motion to see whether we are able to trap cells in 3D.

Reviewers' Comments:

Reviewer #1:

Remarks to the Author:

The authors have clearly answered the questions I raised in my review. These were all relatively minor issues in the first place, but I think that the answers provide additional clarity to the paper. Hence, I think the authors for their detailed responses.

Reviewer #2:

Remarks to the Author:

The authors provided detailed responses to most of the points raised

There are three points that I still would like to see corrected

1. Viability - the authors can not argue that previously cell viability has been tested for these frequencies and powers, and at the same time state that the major technical challenge in performing this work was (i) increasing the frequency by an order of magnitude and (ii) dramatically increasing the required acoustic power to apply significant forces (which they estimate as 200pN). I would agree that these are large forces to exert on a cell - perhaps a hundred times those exerted by optical tweezers. If the authors did not test the viability they should state so.

2. Selecting a cell and manipulating it is not cell-selective manipulation. Cell selective manipulation is moving a cell relative to other cells, based upon a difference in its form or phenotype. The authors obfuscate on this and their use of the term is extremely mis-leading - especially when used in the title.

3. The authors state that the novelty is their ability "to select a single cell, move it independently of neighboring particles and position it". Although they cite Huang extensively they do not include his PNAS publication which showed "3D acoustic tweezers, which can trap and manipulate single cells and particles along three mutually orthogonal axes of motion" using "3D acoustic tweezers to pick up single cells, or entire cell assemblies, and deliver them to desired locations to create 2D and 3D cell patterns, or print the cells into complex shapes" see Proceedings of the National Academy of Sciences Feb 2016, 113 (6) 1522-1527; DOI: 10.1073/pnas.1524813113. The novelty of the work should be discussed in the context of this paper.

Otherwise I am happy to see this published.

Reviewer #3:

Remarks to the Author:

The authors have carefully examined the points raised in the comments, and have modified the manuscript accordingly. I suggest publishing the manuscript as is.